# Temporal Characterization of the Amyloidogenic APPswe/PS1dE9;hAPOE4 Mouse Model of Alzheimer’s Disease

**DOI:** 10.3390/ijms25115754

**Published:** 2024-05-25

**Authors:** Martine B. Grenon, Maria-Tzousi Papavergi, Praveen Bathini, Martin Sadowski, Cynthia A. Lemere

**Affiliations:** 1Ann Romney Center for Neurologic Diseases, Brigham and Women’s Hospital, Harvard Medical School, Boston, MA 02115, USA; mgrenon1@bwh.harvard.edu (M.B.G.); mtzousipapavergi@bwh.harvard.edu (M.-T.P.); pbathini@bwh.harvard.edu (P.B.); 2Section Neuropsychology & Psychopharmacology, Faculty of Psychology and Neuroscience, Maastricht University, 6229 ER Maastricht, The Netherlands; 3Department of Psychiatry and Neuropsychology, School for Mental Health and Neuroscience (MHeNs), Maastricht University, 6200 MD Maastricht, The Netherlands; 4Departments of Neurology, Psychiatry, and Biochemistry and Molecular Pharmacology, New York University Grossman School of Medicine, New York, NY 10016, USA; marcin.sadowski@nyulangone.org

**Keywords:** Alzheimer’s disease, apolipoprotein E, amyloid-related imaging abnormalities (ARIAs), cerebral amyloid angiopathy, cholesterol, human *APOE*-targeted replacement mice

## Abstract

Alzheimer’s disease (AD) is a devastating disorder with a global prevalence estimated at 55 million people. In clinical studies administering certain anti-beta-amyloid (Aβ) antibodies, amyloid-related imaging abnormalities (ARIAs) have emerged as major adverse events. The frequency of these events is higher among apolipoprotein ε4 allele carriers (*APOE4*) compared to non-carriers. To reflect patients most at risk for vascular complications of anti-Aβ immunotherapy, we selected an APPswe/PS1dE9 transgenic mouse model bearing the human *APOE4* gene (APPPS1:E4) and compared it with the same APP/PS1 mouse model bearing the human *APOE3* gene (*APOE* ε3 allele; APPPS1:E3). Using histological and biochemical analyses, we characterized mice at three ages: 8, 12, and 16 months. Female and male mice were assayed for general cerebral fibrillar and pyroglutamate (pGlu-3) Aβ deposition, cerebral amyloid angiopathy (CAA), microhemorrhages, apoE and cholesterol composition, astrocytes, microglia, inflammation, lysosomal dysfunction, and neuritic dystrophy. Amyloidosis, lipid deposition, and astrogliosis increased with age in APPPS1:E4 mice, while inflammation did not reveal significant changes with age. In general, *APOE4* carriers showed elevated Aβ, apoE, reactive astrocytes, pro-inflammatory cytokines, microglial response, and neuritic dystrophy compared to *APOE3* carriers at different ages. These results highlight the potential of the APPPS1:E4 mouse model as a valuable tool in investigating the vascular side effects associated with anti-amyloid immunotherapy.

## 1. Introduction

Alzheimer’s disease (AD) is an irreversible brain disorder clinically defined by the progressive loss of cognitive function, leading to dementia and, ultimately, death [1,2]. Nearly 55 million people worldwide live with dementia today, and it is predicted that by 2050, nearly 13 million will be living with AD in the United States of America [3,4]. The annual cost of care for patients with dementia is becoming one of the biggest economic strains on health-care systems and communities worldwide [4,5].

Immunotherapy is suggested as the most promising AD disease-modifying treatment approach, and recent studies with amyloid-β (Aβ)-plaque-binding monoclonal antibodies (mAbs) have demonstrated plaque lowering and moderate cognitive stabilization in mice [6,7,8,9,10,11] and in humans [10,12,13,14,15]. Two plaque-clearing mAbs, lecanemab and donanemab, showed clinical benefits compared to placebo during their phase 3 trials and received accelerated U.S. Food and Drug Administration (FDA) approval [16,17]. Both mAbs have begun entering clinical practice for patients with mild cognitive impairment (MCI) or mild dementia due to AD. In July of 2023, the FDA converted Leqembi (lecanemab-irmb) to traditional approval following a confirmatory trial that verified its clinical benefit [1].

A caveat of antibodies in this class is highlighted on their prescribing information: a warning about amyloid-related imaging abnormalities (ARIAs). These vascular inflammatory adverse events, observed on magnetic resonance imaging (MRI) scans, have been reported in clinical trials administrating certain anti-Aβ-plaque-binding antibodies [18,19,20]. While the cause of ARIAs is unknown, it has been suggested that Aβ clearance by anti-amyloid antibodies is mediated by perivascular drainage, which may transiently lead to amyloid accumulating in blood vessel walls and inducing inflammation, which, in turn, may increase blood–brain barrier (BBB) breakdown, leading to edema (ARIA-E) or microhemorrhages (ARIA-H) [20,21]. 

ARIA incidence appears to be both dose- and apolipoprotein E (*APOE*)-genotype-dependent and especially pervasive in individuals with CAA [19,22]. ApoE, a polymorphic glycoprotein primarily produced by astrocytes in the brain, was originally discovered as a protein associated with various plasma lipoproteins [23]. Impaired cholesterol metabolism has been proposed to contribute to AD [24]. Cholesterol is released after its conversion to oxysterols or as an apolipoprotein-containing lipoprotein complex. Lipoproteins are composed of cholesterol, triglycerides, and various apolipoproteins. ApoE is primarily responsible for lipid transport and the maintenance of cholesterol homeostasis, as well as mediating the clearance of plasma lipoproteins (by acting as a ligand for lipoprotein uptake by low-density lipoprotein (LDL) and the LDL receptor). Additionally, apoE-containing lipoproteins support the redistribution of cholesterol to cells requiring reparative processes, including injured and regenerating neurons [25,26,27]. Humans have three versions of the *APOE* gene: *APOE* ε2 (*APOE2*), *APOE* ε3 (*APOE3*), and *APOE* ε4 (*APOE4*) alleles [28,29]. ApoE influences Aβ metabolism, conformation, aggregation, and clearance, as well as toxicity, without having a direct effect on the production of Aβ peptides [30,31,32,33,34,35,36,37]. The discovery of the colocalization of apoE with Aβ plaques in the early 1990s [37,38] was closely followed by the revelation that the *APOΕ4* allele is genetically linked to the incidence of AD [39,40]. Notably, the risk of AD has been shown to be 2–3-fold higher in people with one *APOE4* allele and about 12–15-fold higher in those with two alleles [39,41]. The *APOE4* allele also confers a significant risk of ARIAs, where the frequency of ARIA events is substantially higher among *APOE4* allele carriers compared to non-carriers [42]. In APP transgenic mice, the *APOE4* isoform exhibits an impaired ability to induce Aβ proteolysis compared to other *APOE* isoforms, resulting in a higher amyloid load [43]. 

With the hope of comparing the patient risk for vascular adverse events in anti-Aβ immunotherapy, we selected an AD transgenic mouse model (APPswe/PS1dE9 [44,45,46,47]) in which murine *APOE* has been replaced by human *APOE4* or human *APOE3*, and its expression remains controlled by the native murine *APOE* promoter [48,49]. Previous studies have shown that the expression of human *APOE4* in mice producing the wild-type human Aβ peptide leads to a higher propensity to develop CAA [50,51]. It has been demonstrated that once Aβ fibrillogenesis occurs in mice expressing human *APOE4*, there is a shift in Aβ deposition from the brain parenchyma to the vasculature led by *APOE4* [50]. Furthermore, it has been noted that, in AD transgenic mice, the isoform-specific effects of human apoE on Aβ levels and neuritic plaque formation mimic those seen in patients with AD (E4 > E3 > E2), as observed in human post-mortem autopsy and Aβ PET studies [33,52,53,54].

The target replacement mouse lines, hereafter designated as APPPS1:E4 and APPPS1:E3, have not yet been extensively characterized in terms of the temporal progression of amyloid pathology. We analyzed the histology and biochemistry of these mouse models across three ages in both female and male mice to better understand the pathological changes that occur during the development of AD pathology and to determine their relevance as mouse models to study anti-amyloid-antibody-induced ARIAs.

## 2. Results

### 2.1. Aβ Deposition

General Aβ plaque deposition in the prefrontal cortex (PFC) and hippocampus (HC) of the APPPS1:E4 and APPPS1:E3 mouse models was evaluated by immunostaining with S97, a general polyclonal antibody recognizing multiple epitopes within the N-termini of Aβ fragments [55]. Thioflavin-S (ThS) fluorescent staining was used to identify fibrillar Aβ plaques in the hippocampal area. Mice at different ages were used to define the age of onset of Aβ aggregation and determine differences in Aβ deposition during aging. Female and male mice were utilized to determine whether significant sex differences appear in plaque formation in the cortex and hippocampus in these mice. Additionally, the usage of both APPPS1:E4 and APPPS1:E3 mice enabled us to compare the genotype effect on Aβ deposition and aggregation, with a particular focus on CAA. There were significant interaction effects between sex and age in the deposition of fibrillar Aβ (ThS; Figure 1B) and general Aβ (S97; Figure 1E,H). Only in 16-month-old mice did female APPPS1:E4s display significantly more general Aβ (HC: *p* < 0.0001; PFC: *p* < 0.05) and fibrillar Aβ (HC: *p* = 0.0027) deposition than males (Appendix A). A genotype effect was also only detected at the oldest age. Sixteen-month-old mice carrying the *APOE4* allele showed a significant increase in the deposition of general Aβ (Figure 1F, HC: *p* < 0.0001; Figure 1I, PFC: *p* < 0.0001) and fibrillar Aβ (Figure 1C, HC: *p* < 0.0001) compared to APPPS1:E3 mice. We also assessed the concentrations of major Aβ isoforms in the guanidine-soluble (“insoluble”) fractions of hemibrain homogenates. Within APPPS1:E4 mice, there was a significant main effect of age (*p* < 0.0001), where 16-month-old APPPS1:E4 had significantly higher levels of insoluble Aβ40 (*p* < 0.0001) and Aβ42 (*p* < 0.0001) than at other age points. Consistent with general and fibrillar Aβ immunoreactivity, 16-month-old female APPPS1:E4 mice displayed significantly higher levels of Aβ40 (*p* = 0.0139) than their male counterparts (Appendix A). Once more, we saw a significant interaction between age and genotype (Aβ40: *p* < 0.0001; Aβ42: *p* = 0.0001). The effect of the *APOE4* allele was only seen at 16 months, where APPPS1:E4 mice had significantly higher levels of Aβ40 (Figure 1J, *p* < 0.0001) and Aβ42 (Figure 1K, *p* < 0.0001). Overall, in the APPPS1:E4 mice, Aβ42 was the most abundant species at all ages, and the ratio of Aβx-42 to Aβx-40 decreased with age (Figure 1L, *p* < 0.0001). This indicated early high levels of Aβx-42, which equilibrated with Aβx-40 levels by 16 months. APPPS1:E3 mice exhibited a significantly larger ratio value in comparison to APPPS1:E4 mice (Figure 1M), suggesting that at 12 (*p* = 0.0203) and 16 months (*p* = 0.0005), APPPS1:E3 showed higher relative Aβ42 levels than Aβ40. In summary, Aβ pathology in APPPS1:E4 mice seemed to be more pronounced at 16 months of age in both hippocampal and cortical areas, with a more severe emerging pathology in female mice. 

Pyroglutamate-modified Aβ (pGlu-3 Aβ) is one of the most prominent truncated and post-translationally modified peptides in the AD brain and accumulates over time. This peptide is detected in the human brain only under pathological conditions [56], hastens the aggregation of other Aβ peptides in a prion-like manner [57], and shows enhanced stability [58]. These features make pGlu-3 Aβ a highly attractive alternative target for AD immunotherapy, which is why we examined its deposition in this study. K17, a pGlu-3 Aβ IgG2b monoclonal antibody, was used to measure cerebral pGlu-3 Aβ deposition. APPPS1:E4 mice developed pGlu-3 Aβ plaques in the hippocampus and prefrontal cortex as mice aged, with significant deposition at 16 months (Figure 2B, HC: *p* < 0.0001; Figure 2E, PFC: *p* < 0.0001), albeit much lower in abundance than more general Aβ deposition. A significant sex effect was observed at 16 months of age for the prefrontal area in the APPPS1:E4 mice, where females had a higher predisposition to pGlu3 Aβ aggregates (Appendix A, *p* = 0.0334). When considering both genotype groups, we found significant differences in K17 immunoreactivity. In the hippocampal region, at 8 and 12 months of age, APPPS1:E3 mice had nearly undetectable levels of pGlu-3 Aβ (*p* = 0.9862). At 16 months, pGlu-3 Aβ deposition was significantly higher in APPPS1:E4 vs. APPPS1:E3 mice (Figure 2C, *p* = 0.0001). In the prefrontal cortex, a significant interaction effect of the *APOE* genotype and age was found (*p* < 0.0001). Only at 16 months of age was pGlu-3 Aβ deposition significantly higher in APPPS1:E4 vs. APPPS1:E3 mice (Figure 2F, *p* < 0.0001). PGlu-3 Aβ levels were also quantitatively determined by ELISAs on insoluble whole-hemibrain homogenates. In APPPS1:E4 mice, pGlu-3 Aβ levels significantly rose with age (Figure 2G, *p* < 0.0001). Sex differences were recapitulated at 16 months, with females having significantly higher pGlu-3 Aβ levels in the prefrontal cortex (Appendix A). Notably, similar levels of insoluble pGlu-3 Aβ were seen between APPPS1:E3 and APPPS1:E4 mice (Figure 2H, *p* = 0.1982). 

### 2.2. CAA and Microhemorrhage 

In CAA, Aβ is mainly deposited as Aβ fibrils with a characteristically patchy and segmental distribution [59]. Accordingly, the whole hemibrain (including cerebellum) was assessed for positive fibrillar vascular Aβ (ThS). At 16 months, APPPS1:E4 mice showed a significant increase in vascular amyloid compared to younger ages (Figure 3B, *p* = 0.0046). While there was no main effect of genotype when analyzing both the APPPS1:E3 and APPPS1:E4 mice, a slight trend of increased vascular amyloid was visible at 8 and 16 months for APPPS1:E4 mice (Figure 3C). Hemosiderin staining was used to qualitatively investigate the presence of microhemorrhages. The number of hemosiderin-positive iron deposits in whole sagittal hemibrain sections was quantified and averaged for 12- and 16-month APPPS1:E3 and APPPS1:E4 mice (Figure 3E). Hemosiderin deposits frequently occurred in APPPS1:E4 mice, and their number did not increase with age (*p* = 0.8974). In effect, the 16-month-old mice exhibited a slightly lower mean number of deposits than the 12-month-old mice, but this difference was not significant. The main effect of genotype was not quite significant (*p* = 0.0814); however, a trend of increased hemosiderin staining in APPPS1:E4 is visualized in Figure 3E, as APPPS1:E3 mice exhibited little spontaneous microhemorrhage compared to APPPS1:E4 mice.

### 2.3. Lipid Composition

To explore the effects of the *APOE* genotype and age on apoE expression levels, we stained both the hippocampal and frontal cortex areas with an antibody reactive to endogenous total apoE protein and overexpressed human apoE3 and apoE4 proteins. In the hippocampus of APPPS1:E4 mice, there were no sex-dependent differences in apoE immunoreactivity, but there was a main effect of age (*p* < 0.0001). Sixteen-month-old APPPS1:E4 mice had significantly increased levels in comparison to their younger counterparts (Figure 4B). In contrast, APPPS1:E3 mice had very low expression levels, nearing non-measurable at 16 months. We found an interaction between the *APOE* allele and age (*p* = 0.0007), where APPPS1:E3 mice had significantly lower levels of apoE reactivity at 16 months of age (Figure 4D, *p* < 0.0001). In the frontal area of the cortex, the analysis of APPPS1:E4 mice showed an interaction between age and sex (*p* = 0.0356). Sixteen-month-old APPPS1:E4 females had significantly higher positive apoE immunoreactivity in comparison to their male counterparts (*p* = 0.0376). Only within the females was there an age effect, where 16-month-old female APPPS1:E4 mice had significantly higher levels of apoE in comparison to females in other age groups (Figure 4E). When evaluating the frontal area of the cortex in 16-month-old mice, APPPS1:E4 mice had significantly higher apoE immunoreactivity than APPPS1:E3 mice (Figure 4F, *p* = 0.0035). We measured apolipoprotein levels in the guanidine hydrochloride-soluble brain fractions. APPPS1:E4 mice exhibited a main effect of age (*p* = 0.0003), with 16-month-old mice displaying significantly higher levels of apoE than the two younger age groups (Figure 4S). There was no main effect of the *APOE* allele when evaluating both the APPPS1:E3 and APPPS1:E4 mice (Figure 4T, *p* = 0.2003). Mice with the *APOE4* allele appeared to have higher apoE levels at 12 and 16 months, but the difference was insignificant. To assess the co-expression of apoE and Aβ, we utilized a fluorescent amyloid-specific histochemical tracer, Amylo-Glo. We found consistent co-deposition of Aβ and apoE in the hippocampus and frontal cortex of APPPS1:E4 mice. In both brain regions, colocalization increased significantly at 16 months of age in APPPS1:E4 mice (Figure 4G,I). When considering the *APOE* allele, there was an interaction effect of genotype and age in both brain areas (HC: *p* = 0.0003; FC: *p* = 0.0021). Here, the effect of the *APOE* allele was only seen at 16 months of age, where APPPS1:E4 mice demonstrated significantly increased colocalization when compared to the *APOE3* carriers (Figure 4H, HC: *p* < 0.0001; Figure 4J, FC: *p* = 0.0005). Next, we examined the levels of total cholesterol, HDL cholesterol, and triglycerides in the blood plasma. A colorimetric analysis of total cholesterol in APPPS1:E4 mice revealed no main effects of age, sex, or genotype (Appendix A). While no differences were seen across ages in the triglyceride measurements of APPPS1:E4 mice, there was a main effect of sex (*p* = 0.0050), with a trend of males showing higher levels of triglycerides compared to females, but this was insignificant in multiple-comparison tests. APPPS1:E3 and APPPS1:E4 mice had similar triglyceride levels (Appendix A, *p* = 0.3265). No significant main effects of age, sex, or genotype were found in HDL-cholesterol-level analyses for APPPS1:E3 and APPPS1:E4 mice (Appendix A).

### 2.4. Astrogliosis

Glial fibrillary acidic protein (GFAP), an intermediate filament, is a major component of the astrocyte cytoskeleton. Increased GFAP expression is widely used as a marker for astrogliosis [60]. Triple immunofluorescent staining with Amylo-Glo, a GFAP antibody, and an apoE antibody was utilized to visualize reactive astrocytes in tissue sections. In APPPS1:E4 mice, there were no sex or age differences in GFAP immunoreactivity in the hippocampus (Figure 4K). The analysis of hippocampal GFAP reactivity in APPPS1:E3 and APPPS1:E4 mice showed no significant main effect of genotype (Figure 4L, *p* = 0.1305). However, APPPS1:E3 mice displayed a significant increase in astrogliosis in the hippocampus from 8 to 16 months of age (*p* = 0.0417). In contrast, in the frontal cortex of APPPS1:E4 mice, there were main effects of age and sex. At 16 months, females had significantly more GFAP than males (*p* = 0.0027). Sixteen-month-old male APPPS1:E4 mice exhibited significantly increased reactivity in comparison to the other two age groups, and female APPPS1:E4 mice had a significant increase in astrogliosis with age (Figure 4M). Further analysis of both genotypes uncovered an interaction between the *APOE* allele and age (*p* = 0.0263). *APOE4* carriers had significantly higher GFAP reactivity than *APOE3* carriers at 16 months of age (Figure 4N, *p* = 0.0004). APPPS1:E3 mice had greater mean GFAP-positive immunoreactivity in the hippocampal region in comparison to the frontal cortex area. Plaque-associated reactive astrocytes were quantified by Amylo-Glo and GFAP colocalization. In both regions of the brain, there were no main effects of age or sex. Nevertheless, there was a main effect of genotype in both the hippocampus (*p* = 0.0159) and frontal cortex (*p* = 0.0101). In the hippocampus, APPPS1:E4 mice showed a non-significant trend of increased colocalization of amyloid plaques and astrocytes in comparison to APPPS1:E3 mice (Figure 4O). This trend was replicated in the frontal cortex, albeit significant at 8 months of age (Figure 4P, *p* = 0.0209). As apoE is primarily produced by astrocytes [61], we explored the colocalization of apoE and GFAP reactivity. Overall, there were no sex differences in APPPS1:E4 mice. In the hippocampus, colocalization increased significantly with age, where 16-month-old mice exhibited significantly more overlap between GFAP- and apoE-positive staining than younger mice (*p* < 0.0001). In the frontal cortex, there was only a significant increase from 8 to 16 months of age (*p* = 0.0235). In the hippocampus, there was an interaction effect of genotype and age with regard to GFAP and apoE colocalization (*p* < 0.0001). At 16 months of age, APPPS1:E4 mice had significantly more apoE and GFAP colocalization compared to APPPS1:E3 mice (Figure 4Q). However, in the frontal cortex, there was only a significant main effect of genotype (*p* = 0.0486). Similar to the hippocampus, APPPS1:E3 mice had significantly reduced apoE and GFAP colocalization at 16 months of age (Figure 4R, *p* = 0.0286). 

### 2.5. Inflammatory Markers

Next, we analyzed the blood plasma of APPPS1:E3 and APPPS1:E4 mice for inflammatory biomarkers. The panel included the following 10 analytes involved in the inflammatory response and immune system regulation: interferon-gamma (IFN-γ), interleukin (IL)-1β, IL-2, IL-4, IL-5, IL-6, IL-10, IL-12p70, KC/GRO (IL-8-related protein in rodents), and tumor necrosis factor-alpha (TNF-α). Due to low plasma volumes, samples from both sexes were pooled, and sex differences were not considered. The levels of IL-4 and IL-12p70 were below the level of detection in all groups. There were no main effects of age in the APPPS1:E4 mice. Non-significant trends were seen in APPPS1:E4 mice: IL-1β (Figure 5A, *p* = 0.1215), IL-6 (Figure 5E, *p* = 0.1215), and TNF-α (Figure 5K, *p* = 0.1086) levels increased with age, and KC/GRO (Figure 5I, *p* = 0.0578) levels declined with age. There were no differences in IFN-γ, IL-2, or IL-10 levels in APPPS1:E4 mice (*p* > 0.5). The analysis of both APPPS1:E3 and APPPS1:E4 mice showed significant main effects of age (*p* = 0.0110) and genotype (*p* = 0.0257) on TNF-α levels. Sixteen-month-old APPPS1:E3 mice had significantly higher levels of TNF-α than 12-month-old APPPS1:E3 mice (*p* = 0.0434). When considering the effect of the *APOE* allele, there was a trend for 12-month APPPS1:E4 mice to have higher TNF-α levels than APPPS1:E3 mice (Figure 5L, *p* = 0.1171). Although non-significant, there was a trend for a genotype effect on KC/GRO levels (Figure 5J, *p* = 0.0669). Other markers had no main effects on genotype. Unpaired *t*-tests of IL-5 levels were significant at 8 (Figure 5B, *p* = 0.0234) and 12 months (Figure 5C, *p* = 0.0043) of age, and APPPS1:E4 mice had higher levels than APPPS1:E3. Furthermore, slightly higher IL-6 levels were seen in APPPS1:E4 mice at 12 months (Figure 5G, *p* = 0.1120). 

### 2.6. Microglial Response

Ionized calcium-binding adaptor molecule 1 (Iba1) is a calcium-binding protein located in microglia and macrophages and is widely utilized as a microglial/macrophage marker [62]. Cluster of differentiation 68 (CD68), a member of the lysosomal/endosomal-associated membrane glycoprotein (LAMP) family, is a transmembrane glycoprotein and serves as a marker of the phagocytic activity of microglia and macrophages [63,64]. Triple immunofluorescence staining was performed with Amylo-Glo dye and Iba1 and CD68 antibodies to visualize phagocytic microglia (CD68 and IBA1) and the amount of amyloid located within lysosomes (plaque-associated lysosomal activity; CD68 and Amylo-Glo) in the brains of APPPS1:E4 and APPPS1:E3 mice. Female and male mice were utilized to determine whether significant sex differences appear in the microglial response and lysosomal activity during aging. Female APPPS1:E4 mice seemed to present higher values of phagocytic microglia and plaque-associated lysosomal activity than males throughout aging; however, the statistical analysis of the sex effect data showed no significance, and therefore, these data are not presented here. 

Representative microphotographs of the hippocampus and frontal cortex of APPPS1:E4 and APPPS1:E3 (8 months) mice in each color channel of interest, as well as with all three channels merged together to visualize colocalization, can be found below (Figure 6A,F,G). Quantitative image analysis was performed by quantifying the percentage colocalization area of CD68 and Iba1 staining and CD68 and Amylo-Glo staining to identify phagocytic microglia and plaque-associated lysosomal activity, respectively. Phagocytic activity was significantly higher in APPPS1:E4 mice compared to APPPS1:E3 ones at 8 months of age in both the hippocampus (*p* < 0.05) and frontal cortex (*p* < 0.01) (Figure 6B,D). Plaque-associated lysosomal activity did not show any significant differences between genotype groups in 8-month-old mice; nevertheless, it was slightly higher in APPPS1:E4 mice in both brain areas of interest (Figure 6C,E). Phagocytic microglia and plaque-associated lysosomal activity did not show any significant differences between APPPS1:E4 and APPPS1:E3 mice at 12 and 16 months of age, and thus, these data are not presented here. APPPS1:E4 mice were not characterized by significant changes in phagocytic activity during aging in either the hippocampus or frontal cortex (Figure 6H,I). An increase was detected at 12 and 16 months compared to 8 months, but it still did not reach statistical significance. Therefore, aging does not seem to significantly affect the phagocytic activity of microglia in APPPS1:E4 mice. Likewise, APPPS1:E4 mice did not show significant alterations in plaque-associated lysosomal activity during aging in either the hippocampus or frontal cortex (Figure 6J,K). Notably, plaque-associated lysosomal activity decreased with age in the hippocampal area (Figure 6J), with the lowest levels detected at 16 months of age. In the frontal cortex, the amount of amyloid in lysosomes was similar between 8- and 12-month-old mice, while, once again, it was the lowest at 16 months of age (Figure 6K). These results suggest that APPPS1:E4 mice had an impaired response to plaque deposition throughout aging.

### 2.7. Lysosomal Dysfunction and Neuritic Dystrophy

Lysosomal-Associated Membrane Protein 1 (LAMP1) is a protein that is mainly found in the lysosomal membrane and is involved in the maintenance of structural integrity in lysosomes. LAMP1^+^ immunoreactivity is present in dystrophic neurites within neuritic plaques in AD [65]. LAMP1 was used here as a marker of lysosomal dysfunction. Sternberger monoclonals incorporated (SMI)-31 is an anti-Neurofilament H monoclonal antibody specifically targeting phosphorylated epitopes on the heavy subunit of neurofilament proteins. Neurofilaments are crucial to neuronal integrity [66], and SMI-31 was utilized in this study to detect dystrophic neurites. Triple immunofluorescence staining was performed with Amylo-Glo dye and LAMP1 and SMI-31 antibodies to visualize plaque-associated LAMP1^+^ immunoreactivity (LAMP1 and Amylo-Glo), proteasomal dysfunction in neurites (LAMP1 and SMI31), and plaque-associated neuritic dystrophy (SMI-31 and Amylo-Glo) in the brains of APPPS1:E4 and APPPS1:E3 mice. Female and male mice were utilized to determine whether significant sex differences appear in lysosomal dysfunction and neuritic dystrophy during aging. Female APPPS1:E4 mice had increased levels of plaque-associated LAMP1^+^ immunoreactivity (Appendix A) and neuritic dystrophy (Appendix A), as well as increased proteasomal dysfunction in neurites (Appendix A), compared to males at 16 months of age. Hence, 16-month-old female APPPS1:E4 mice had more neuritic plaques than males at the same age. Male APPPS1:E4 mice presented higher values of plaque-associated LAMP1^+^ immunoreactivity (Appendix A) and proteasomal dysfunction in neurites (Appendix A) than females at 8 and 12 months of age. Both sexes had undetectable levels of plaque-associated neuritic dystrophy at 8 and 12 months of age.

Representative microphotographs of the hippocampus and frontal cortex of APPPS1:E4 mice and the frontal cortex of APPPS1:E3 mice in each color channel of interest, as well as with all three channels merged together to visualize colocalization, can be found below (Figure 7A). Data from the hippocampi of APPPS1:E3 mice were characterized by high variability and a number of outliers, preventing the proper interpretation of these results. Thus, representative microphotographs of these brains are not included here. Quantitative image analysis was performed by quantifying the percentage colocalization area of LAMP1 and Amylo-Glo staining and SMI-31 and Amylo-Glo staining to identify plaque-associated LAMP1^+^ immunoreactivity and plaque-associated neuritic dystrophy, respectively, as well as the percentage colocalization area of LAMP1 and SMI31 to detect proteasomal dysfunction in neurites. The percentage area of LAMP1^+^ and SMI31^+^ immunoreactivity alone was also quantified to determine the accumulation of lysosomes and dystrophic neurites, respectively, throughout aging and across genotypes. APPPS1:E4 mice had increased levels of lysosome accumulation in both the hippocampus and frontal cortex (Figure 7B,C). In the frontal cortex, this increase seemed to be more prominent throughout aging, characterized by a significant difference between 8 and 16 months (*p* < 0.05; Figure 7C). Additionally, APPPS1:E4 mice had higher levels of lysosomal accumulation at 8 and 16 months of age compared to APPPS1:E3 mice (Figure 7D). Plaque-associated LAMP1^+^ immunoreactivity in APPPS1:E4 mice was unaltered throughout aging in both the hippocampus and frontal cortex (Figure 7E,F). A non-significant increase in plaque-associated LAMP1^+^ immunoreactivity was seen at 16 months of age, with a more pronounced associated immunoreactivity presented in the frontal cortex (Figure 7F) compared to the hippocampus (Figure 7E). Moreover, 12-month-old mice showed a slight decrease in plaque-associated LAMP1^+^ immunoreactivity in comparison to 8-month-old mice in both brain regions (Figure 7E,F). Aging did not have a significant effect on proteasomal dysfunction in neurites in either brain area (Figure 7G,H). However, similar to LAMP1 and Amylo-Glo colocalization, there was a slight decrease in proteasomal dysfunction in neurites at 12 months and an increase at 16 months in both brain areas (Figure 7G,H). Aging did not affect the accumulation of dystrophic neurites in the hippocampus of APPPS1:E4 mice (Figure 7I). On the other hand, a significant increase (*p* < 0.05) in dystrophic neurite accumulation in the frontal cortex in 16-month-old mice compared to 12-month-old mice was observed (Figure 7J). APPS1:E4 mice had higher levels of dystrophic neurites than APPPS1:E3 mice at all age points (Figure 7K). Plaque-associated neuritic dystrophy seemed to start developing in APPPS1:E4 mice at the late age of 16 months in both the hippocampus and frontal cortex (Figure 7L,M). No plaque-associated neuritic dystrophy was detected at the ages of 8 and 12 months in APPPS1:E4 mice. For APPPS1:E3 mice, plaque-associated neuritic dystrophy was hardly detectable at all age points. To summarize, the accumulation of lysosomes and dystrophic neurites was higher in APPPS1:E4 mice at 16 months of age. Aging did not have a strong effect on plaque-associated LAMP1^+^ immunoreactivity or proteasomal dysfunction in neurites in APPPS1:E4 mice, while plaque-associated neuritic dystrophy in this model was present solely in older mice. 

## 3. Discussion

AD is the leading cause of neurodegenerative dementia in elderly people, gravely affecting the quality of life of AD patients but also imposing a great psychological and economic burden on their caretakers. The World Health Organization has declared AD a public health priority [3]. Hence, throughout the years, great effort has been made to shed light on the underlying pathology of this disease and find treatments to slow or stop its progression. Evidence for the amyloid-cascade hypothesis in AD pathogenesis has led to the development of anti-amyloid immunotherapy as a disease-modifying therapy for AD. Animal models serve a critical role in investigating AD pathogenesis and the mechanisms of the disease and in exploring the safety and efficacy of novel therapeutic interventions. To date, there are at least 216 AD rodent models, in addition to non-mammalian and invertebrate models (www.alzforum.org; accessed on 12 August 2023). Transgenic mice overexpressing human mutant APP alone or with PSEN1 on a human *APOE* background have emerged as a promising approach to studying its implications for AD pathogenesis, prevention, or treatment [67]. The proper characterization of a mouse model is necessary for the translational potential of animal studies. Here, we wished to model patients who are most at risk for vascular complications of anti-Aβ immunotherapy, and therefore, we chose APP/PS1 transgenic mouse models bearing the human *APOE4* gene, a strong risk factor for anti-amyloid antibody-induced ARIAs, for comparison with those bearing the human *APOE3* gene. 

Herein, we describe evidence of age-, sex-, and allele-related increases in both general Aβ deposition and in pyroGlu-3-modified Aβ species in both mouse models. Within the hippocampus and prefrontal cortex, general Aβ levels significantly increased with age. Levels of insoluble Aβ40 and Aβ42 in the guanidine fraction also increased with age, although this result was only significant at 16 months of age. As anticipated based on previous studies in humans and transgenic mice, the presence of a human *APOE4* allele is associated with higher Aβ levels [52,68,69,70,71]. At 16 months of age, a genotype-related difference in Aβ burden was statistically significant. Within the clinical realm, a decrease in the ratio of plasma Aβ42 to Aβ40 has been suggested to be detectable in early disease progression and could therefore serve as a biomarker of Alzheimer’s disease [72,73]. In the brain, compared to Aβ42, the Aβ40 levels in APPPS1:E4 mice rose slowly and at lower levels, initially displaying a higher Aβ42/Aβ40 ratio at 8 and 12 months but reaching equilibrium at 16 months. Intriguingly, the Aβ42/Aβ40 ratio of APPPS1:E3 mice is similar to that of APPPS1:E4 mice at 8 months of age but is significantly larger at 12 and 16 months, suggesting potentially less deposition of Aβ40 within the plaque morphology of APPPS1:E3 mice. This coincides with the general and fibrillar plaque load differences seen between APPPS1:E4 and APPPS1:E3 mice at 16 months of age. It is important to note that the introduction of human *APOE3* and *APOE4* into transgenic mice expressing familial AD mutations has been reported to distinctly delay Aβ deposition in young animals [52,74] and, opposite to humans, is usually observed in older ages [33,50,75], where, once detected, plaque levels are higher with *APOE4* than with *APOE3*, in concordance with our current findings. 

To our knowledge, pGlu-3Aβ pathology has yet to be characterized in this model. Similar to our previous work in other AD-like transgenic mouse models [76], we found that pGlu-3Aβ was lower in abundance and deposited more slowly than general Aβ at all ages in both APPPS1:E4 and APPPS1:E3 mice. While pGlu-3 Aβ levels were low at younger ages, we found a significant increase at 16 months of age. Previous regional differences were affirmed: the cortex, a structure previously shown to have substantial levels of pGlu-3Aβ in other mouse models, had higher pGlu-3 Aβ levels in both genotypes here relative to those in the hippocampus [76,77]. We found inconsistent effects of the *APOE* genotype on pGlu-3Aβ at 16 months. Pathologically, we showed a robust effect of genotype, in which APPPS1:E4 mice had more staining than APPPS1:E3 mice. Similar biochemical levels of insoluble pGlu-3Aβ in the hemibrain homogenates were seen across APPPS1:E3 and APPPS1:E4 mice. We hypothesize two possible caveats: a small sample size, especially for the APPPS1:E3 mice (three mice at 16 months of age), or a possible technical error. Because the staining was successful in the APPPS1:E4 mice, we suspect that the pathological difference may have been due to the small sample size of the APPPS1:E3 mice. 

Previous studies have assessed the fibrillar Aβ plaque load within the APPPS1:E4 mouse model. However, to our knowledge, none have examined this temporally. Our present study confirms age-related increases in fibrillar ThS-positive plaques, with 16-month-old mice presenting higher loads than at younger time points. We extended this observation by showing sex-related changes only seen in 16-month-old mice, wherein females displayed a higher plaque burden. Clinical studies of fibrillar Aβ plaque loads have shown an allele-specific effect, with human *APOE4* allele carriers exhibiting higher Aβ deposition than *APOE3* carriers [78,79,80]. In line with Pankiewicz and colleagues’ 2017 study with 15-month-old vehicle-injected APPPS1:E4 mice, we found that APPPS1:E4 mice exhibited a higher hippocampal fibrillar plaque load than APPPS1:E3 mice at 16 months [81]. Diverging from our findings, Kuszczyk and peers [82] saw an effect of the *APOE* genotype on the fibrillar Aβ load at an earlier time point (11 months). However, their fibrillar Aβ load quantification was within the neocortex of coronal sections and only included female mice [82]. For this investigation, parenchymal and vascular fibrillar Aβ were not differentiated. Future quantification of the cortex may highlight similar modulatory effects. 

The development of immunotherapies against Aβ is burdened by clinical adverse events observed as ARIAs on MRI scans. Vascular inflammatory adverse events, including vasogenic edema and/or cerebral microhemorrhages, have been reported in AD clinical trials upon treatment with anti-Aβ-plaque-binding antibodies. ARIA incidence appears to be *APOE*-genotype-dependent, wherein *APOE4* allele carriers are at higher risk compared to non-carriers. It has been previously suggested that the grade of treatment-related effects is likely to vary depending on the severity of the baseline vascular Aβ pathology [83]. Vascular pathology may play an important role in the etiology of AD by decreasing cerebral blood flow and impairing Aβ clearance [84]. One hypothesis for ARIAs is that prolonged inflammation from Aβ antibodies targeting CAA may impair vascular integrity and lead to ARIAs [85]. Accordingly, we semi-quantitatively characterized ThS-positive deposits of fibrillar Aβ in the cerebral vasculature and hemosiderin deposits, reflecting microhemorrhages. Cerebrovascular amyloid was significantly higher in 16-month-old APPPS1:E4 mice compared to age-matched APPPS1:E3 mice. Prior studies in APP transgenic mice showed an association between an increased risk for CAA and the presence of the *APOE4* allele [42,50,86,87,88,89]. In 12- and 16-month-old mice, hemosiderin labeling of microbleeds frequently occurred in APPPS1:E4 mice, and their number did not vary with age. APPPS1:E3 mice exhibited little spontaneous microhemorrhage compared to the APPPS1:E4 mice. These results are limited by the large variability within the APPPS1:E4 group. Future investigations of BBB leakiness in these mice will offer valuable insights, particularly concerning the potential of the intricate interplay between BBB integrity and neurovascular pathology.

The effect of *APOE* on AD pathophysiology is multifactorial and sometimes even independent of Aβ, including effects on neuroinflammation, lipid metabolism, and the BBB [90]. In the presence of an intact BBB, the human brain is not influenced by circulating cholesterol, and brain cholesterol metabolism is self-contained [24]. It is suggested that caution be taken when considering causal correlations between brain pathology and cholesterol levels, as CNS cholesterol is made locally, with the BBB limiting cholesterol supply from the blood. However, the breakdown of the BBB has been observed as an early sign of AD [91]. In vitro and in vivo, abnormal levels of neuronal cholesterol have been linked to AD-related pathology; however, the cholesterol types and their association with AD are inconsistent and not fully understood [92,93]. It has been suggested that cholesterol is related to AD through vascular and inflammatory changes and not changes in brain cholesterol. The three major isotypes of *APOE* are associated with differences in lipoprotein levels and varying affinities for LDL receptors [94]. The apoE4 protein presents a binding preference for large, triglyceride-rich lipoproteins (very-LDLs, LDLs, and chylomicrons) and is associated with high blood LDL cholesterol [95,96,97]. Among the effects of the *APOE4* allele, we should acknowledge hypercholesterolemia-related vascular defects and the direct toxic effects of *APOE4* on the cerebrovascular system [24]. In the current study, we found no significant effect of the *APOE* genotype on the plasma levels of total cholesterol, HDL cholesterol, or triglycerides at any age point. In contrast, in the plasma of 6–8-month-old mice of the same mouse model, Fitz and colleagues [98] found decreased HDL levels in APPPS1:E4 mice when compared to APPPS1:E3s. In 2017, Pankiewicz et al. published similar total cholesterol levels in vehicle-injected 15-month APPPS1:E3 and APPPS1:E4 mice, mirroring our findings at 16 months [81]. It has been suggested that there are correlations between high LDL cholesterol and low HDL cholesterol in patients with early AD [99,100]. Atherosclerosis, which is thought to contribute to AD pathogenesis, is associated with high LDL-cholesterol and triglyceride levels and low HDL-cholesterol levels. The accumulation of cholesterol in Aβ plaques has been demonstrated in AD patients and APPswe mice [101]. Early work with rodents found a link between high cholesterol levels and AD, where a hypercholesterolemic diet increased the Aβ plaque load, gliosis, and tau hyperphosphorylation [102,103,104,105]. A caveat of ours, and many of these studies, is that these studies investigated blood samples drawn at a single point in time. In a large population-based clinical cohort, Moser and colleagues [93] found an association between high variability in total cholesterol and triglyceride levels and incident AD and AD-related dementias (ADRDs). Variation in lipids is defined as any change in lipid levels over time regardless of direction. In contrast, they found that variability in LDL-cholesterol and LDL-cholesterol levels were not associated with AD/ADRD [93]. 

To our knowledge, no other studies on the APPswe/PS1dE9 AD transgenic mouse model with targeted replacement human *APOE* alleles have quantitively examined apoE deposition temporally. We found an age-dependent increase in apoE deposition in both histological immunoreactivity and biochemistry in APPPS1:E4 mice. This is consistent with the findings of Bales et al. [75], who used human *APOE4*-targeted replacement PDAPP mice and also observed age-dependent increases in insoluble apoE levels. They did not publish results on apoE immunoreactivity [75]. With immunostaining, we found that at 16 months, APPPS1:E4 mice had increased cortical and hippocampal total apoE deposition, whereas APPPS1:E3 mice had nearly undetectable amounts. In the guanidine-soluble fraction of homogenized whole hemibrain, we found no significant differences in total apoE levels between APPPS1:E4 and APPPS1:E3 mice. There is a substantiated correlation between apoE plasma levels and the *APOE* genotype in both humans [106,107] and animal models [75,108,109], with *APOE4* carriers having the lowest protein levels. However, the presence and direction of an *APOE* allele genotype effect on CNS-synthesized apoE protein levels are not consistent. In clinical studies with brain homogenates of AD patients, there is evidence for an *APOE4* genotype-dependent reduction in apoE protein levels [110,111]. Notably, in a 2011 study with brain homogenates from AD patients, comparisons of Western blot densitometry data showed significant increases in the amount of apoE in the brains of AD patients homozygous for *APOE4*. In brain homogenates of *APOE* transgenic and knock-in mouse models, previous studies have found that insoluble homogenate apoE levels were reduced [52], unchanged [50,112,113], and increased [75] in *APOE4* carriers. The caveats of these studies include variations between methods (Western blot, ELISA), fractions utilized (soluble, detergent-soluble vs. insoluble), brain region(s) homogenized, mouse models (genes implicated in AD, *ApoE* promoter), and age. ApoE has been shown to be dramatically upregulated after neuronal injury. Moreover, prior work has suggested that apoE competes with Aβ for cellular uptake [114,115]. It is possible that the abundant deposition of both apoE and Aβ in the brains of 16-month APPPS1:E4 mice is due to the direct competition for the clearance of Aβ, apoE, and/or the apoE-Aβ complex. The polymorphisms of *APOE* alter protein structure and modify its ability to bind to Aβ [116]. The binding of apoE to Aβ is further affected by the apoE lipidation state and the aggregation state of Aβ [114,117]. The majority of studies have found that the efficiency of complex formation between Aβ and lipidated apoE follows the following sequence: apoE2 > apoE3 >> apoE4 [118]. We found that the majority of apoE measured in the hippocampus (60%) and frontal cortex (66%) of the oldest age group of APPPS1:E4 mice was colocalized with Aβ. This colocalization also increased with age, whereas there was only a minor association of apoE with Aβ (20%) in the frontal cortex of APPPS1:E3 mice, suggesting that the rise in apoE deposition observed in the aged APPPS1:E4 mice is likely a result of the deposition of apoE with Aβ.

There are several pathways by which Aβ can be cleared from the brain, including clearance through the BBB, enzymatic degradation, cellular internalization, and subsequent lysosomal degradation by microglia and astrocytes [119]. Astrocytes have a broad range of functions, have been implicated in AD-related inflammatory and pathological processes, and are the primary source of apoE in the brain [42,120,121]. *APOE* can alter the expression profile and functioning of astrocytes, including responses to pathogenic stimuli [119,122,123]. A common feature of astrocytes responding to injury, disease, or infection (reactive astrocytes) is the upregulation of GFAP [121,124]. Habib and colleagues [125] found that a population of GFAP astrocytes, coined “disease-associated astrocytes”, appeared early in AD mouse models and increased in abundance with disease progression. In an alternate model of amyloidosis, the removal of *APOE4* reduced GFAP reactivity [123]. In this study, the levels of the astrocytic protein GFAP were inconsistent across the two brain regions that we investigated. In the frontal cortex, the levels of GFAP were parallel with those of apoE, significantly higher in APPPS1:E4 mice at 16 months, and increased with age. However, in the hippocampus, there was a steady percentage of GFAP-positive astrocytes in both the APPPS1:E3s and APPPS1:E4s. The effect of the *APOE* allele on plaque-associated reactive astrocytes was significant, with *APOE4* mice tending to have higher colocalization of GFAP and fibrillar Aβ in both brain regions, potentially suggesting that astrocytes in the *APOE4*-positive mice were unable to degrade Aβ deposits. We found a significant increase in astrocyte and apoE colocalization in 16-month-old *APOE4* allele carriers in comparison to APPPS1:E3 mice. 

We examined several inflammatory biomarkers in blood plasma that have previously implicated the *APOE4* allele in a greater inflammatory response [126,127,128,129,130,131]. Our data suggests that *APOE4* carrier mice had increased production of pro-inflammatory cytokines, as our data showed trends of increased levels of IL-6 and TNF-α. TNF-α promotes synaptic loss and excitotoxicity and exacerbates amyloidosis, with heightened levels of serum TNF-α already identified in AD [132,133]. Additionally, TNF-α can induce the increased production of IL-6, whose signaling has been shown to be both pro- and anti-inflammatory [134]. An association between IL-6 and AD has previously been suggested, as AD patients display increased IL-6 expression in the CNS and periphery [135]. Notably, only at 8 and 12 months of age did *APOE4* mice have significantly elevated levels of IL-5 compared to *APOE3* mice. IL-5 is a part of the adaptive immunity classification and, as a pro-inflammatory cytokine, regulates innate and acquired immune responses [136,137]. Lins and Borojevic [138] hypothesized that IL-5 has a specific paracrine/autocrine function in astrocytes, maintaining the homogeneity of the activation state in an astrocyte population. IL-5 activates B cells, stimulating innate B-1 cells and inducing homeostatic proliferation, survival, and differentiation, leading to the secretion of antibodies (Immunoglobulins A and M) [136,139]. This suggests that mice with the *APOE4* allele have a dysfunctional immune response at later ages. 

Under stress conditions, microglia can also produce apoE, albeit to a lesser extent than astrocytes [140,141,142,143]. Microglia are the resident phagocytes of the CNS and significantly contribute to the maintenance of neuronal plasticity and synapse remodeling [144]. The accumulation of Aβ acts as a pathological trigger activating microglia, causing them to migrate and initiate immune responses, leading to the secretion of chemokines and pro-inflammatory cytokines [145], while it may also entail the inflammasome-dependent formation of ASC (apoptosis-associated speck-like protein containing a caspase recruitment domain) specks in microglia [146]. Once activated, microglia seem to have a dual role in the progression of AD [147]. They can have a vital role by assisting Aβ clearance [148] and by creating a physical barrier around plaques, with their processes delaying disease development [149], but can also be detrimental, leading to chronic neuroinflammation and the exacerbation of AD pathology [150]. Over the past few years, a large number of studies have explored the interaction between microglia and *APOE*, revealing differential *APOE*-mediated effects on microglia. In human post-mortem sections of the temporal neocortex, no differences were detected in microglial activation between *APOE4* carriers and non-carriers [151]; however, studies in living individuals across the aging and AD spectrum [152], as well as studies in mice and with the use of transcriptomics [92,153,154,155,156], have demonstrated the important contribution of the *APOE4* allele to the microglial response in AD. Notably, Krasemann and colleagues [157] identified *APOE* signaling as the key modulator of the transcriptional shift from homeostatic to neurodegenerative/disease-associated microglia (MGnD) [157]. The Butovsky lab recently highlighted the cell-intrinsic role of *APOE4* in inducing dysfunctional MGnD microglia with an impaired response to neurodegeneration and showed that the microglial deletion of *APOE4* restores MGnD microglia in APP/PS1 mice [156]. Furthermore, it has been reported that *APOE3* lipoproteins are able to elicit a more effective and faster transcriptional and phenotypic microglial response to Aβ than *APOE4* [155]. Overall, *APOE4* microglia seem to maintain a homeostatic signature, and mice bearing the *APOE4* gene are characterized by lower microglial phagocytic activity [156]. 

Our results indicate that plaque-associated lysosomal activity (CD68 and Amylo-Glo colocalization) seems to decrease with age, especially in the hippocampus, meaning that as AD pathology is exacerbated, microglia become less phagocytic, demonstrating their impaired response to neurodegeneration, which is most probably elicited by the presence of the *APOE4* allele. Phagocytic microgliosis (CD68 and IBA1 colocalization) showed a slight, non-significant increase at 12 months of age compared to 8 months, which appears to saturate by 16 months, reflecting, once again, the fact that as the pathology gets worse in APPPS1:E4 mice, microglia become less phagocytic, an observation that needs to be considered when planning future immunotherapeutic studies. Phagocytic activity was significantly higher in APPPS1:E4 mice at 8 months of age compared to APPPS1:E3 ones, while this difference was lost during aging. Nevertheless, APPPS1:E3 mice did not show an increased microglial response compared to APPPS1:E4 at older ages; hence, we cannot conclude that the microglial response is more effective in APPPS1:E3 mice, as shown by Fitz et al. [155]. A limitation here could be the small sample size of APPPS1:E3 mice used in this study. Collectively, our results demonstrate lower levels of phagocytic activity and an impaired microglial response to Aβ clearance during aging in the hippocampus and frontal cortex of APPPS1:E4 mice.

Lysosomal dysfunction and neuritic dystrophy are two important aspects of AD pathology, and they play a significant role in the development and progression of the disease. Lysosomes are membrane-enclosed organelles that contain digestive enzymes capable of breaking down waste products and cellular debris [158]. To this end, compromised lysosomal activity acts as a major player in protein accumulation diseases, including AD [159,160]. Disrupted lysosome function leads to reduced Aβ and tau clearance, triggering a cascade of downstream events such as neuroinflammation, synaptic plasticity deficits, and cellular stress, consequently contributing to neuritic dystrophy and neurodegeneration [65,161,162]. Neuritic dystrophy is characterized by the formation of dystrophic neurites, structures presented as swollen and abnormal neuronal processes that are mainly filled with autophagic vacuoles [163] but also contain lysosomal vesicles, p-Tau filaments, neurofilament proteins, and amyloid precursor protein (APP) [164]. Dystrophic neurites are encountered surrounding neuritic plaques, also known as senile plaques, a hallmark of AD pathology [165]. In summary, it becomes obvious that lysosomal dysfunction and neuritic dystrophy are intertwined processes in AD, with lysosomal dysfunction leading to neuritic dystrophy via the accumulation of endo-lysosomal and autophagic vesicles in dystrophic neurites surrounding amyloid plaques [163]. Furthermore, investigating the formation of neuritic plaques is critical in AD since they seem to be the structures where AD-related proteinopathies colocalize at the cellular level [166]. 

To our knowledge, plaque-associated LAMP1^+^ immunoreactivity (LAMP1 and Amylo-Glo colocalization) has yet to be characterized in the APPPS1:E4 mouse model. Higher levels of LAMP1^+^ immunoreactivity indicate increased lysosomal accumulation, reflecting higher levels of lysosomal dysfunction. The introduction of the presenilin mutation in transgenic APPswe mice has been reported to accelerate lysosomal dysfunction [161], while the *APOE4* allele was indicated to promote endosomal abnormalities in human post-mortem tissue [167]. Additionally, transcriptomic and immunohistochemical studies utilizing *APOE4* mice [112] indicated a direct link between the *APOE4* gene and endosomal/lysosomal dysregulation [168]. In this study, a significant increase in lysosomal dysfunction (LAMP1) in the frontal cortex of 16-month-old APPPS1:E4 mice was observed. Nevertheless, we did not identify statistically significant alterations in plaque-associated lysosomal dysfunction in the hippocampus or frontal cortex of APPPS1:E4 mice during aging, but elevated levels of dysfunction could be observed at 16 months of age in both brain areas of interest, being more pronounced in the frontal cortex. APPPS1:E4 mice at 16 months of age had higher levels of lysosomal dysfunction compared to APPPS1:E3 mice. Notably, it has been reported that microglia activation acidifies lysosomes, leading to fibrillar Aβ clearance [169], and that microglial lysosomal acidification defects may inhibit phagocytosis and autophagy [170]. Hence, the impaired microglial response observed in our APPPS1:E4 model, especially at 16 months of age, could be the cause of the higher levels of lysosomal dysfunction that we observed with aging. 

Neuritic plaques constitute a major characteristic of AD pathology, and the presence of plaque-associated neuritic dystrophy in AD transgenic mice was reported early on [33]. Holtzman and colleagues revealed that *APOE4* exerts a greater effect on the formation of neuritic plaques compared to the *APOE3* allele, while neuritic plaques could be observed by 15 months of age in *APOE4* mice [33]. More recently, the Holtzman team utilized APPPS1:E4 and APPPS1:E3 mice to investigate the effects of *APOE4* in the context of sleep deprivation in AD and identified less phagocytic and plaque-associated microgliosis and greater levels of dystrophic neurites in the presence of the *APOE4* allele compared to *APOE3* [171]. In our study, APPPS1:E4 mice were characterized by significantly higher levels of dystrophic neurites (SMI31) in the frontal cortex at 16 months of age. APPPS1:E4 mice at 16 months of age had more dystrophic neurites compared to APPPS1:E3 mice. Furthermore, proteasomal dysfunction in neurites (LAMP1 and SMI31 colocalization) was increased in the frontal cortex at 16 months of age. Our results agree with what the Holtzman group previously observed; APPPS1:E4 mice are characterized by an impaired microglial response, leading to increased damage to neurites and exacerbated pathology when compared to APPPS1:E3 mice at a later age. Plaque-associated neuritic dystrophy (SMI31 and Amylo-Glo) in APPPS1:E4 mice was only detected at the later age of 16 months, while APPPS1:E3 mice had hardly detectable levels of plaque-associated neuritic dystrophy. Overall, according to our observations, APPPS1:E4 mice are characterized by higher levels of general and plaque-associated lysosomal dysfunction and proteasomal dysfunction in neurites at 16 months of age, while plaque-associated neuritic dystrophy is a feature of the model that can be detected at a later age, after 12 months and closer to 16.

With a specific focus on the risk of vascular complications associated with anti-Aβ immunotherapy and with the understanding that *APOE4* allele carriers are at higher risk when compared to non-carriers, we selected the APPPS1:E4 mouse model. As this model has human *APOE*, it critically improves the translatability of results to patients. In contrast to humans, APPPS1:E4 mice display general Aβ deposition prior to pGlu-3-Aβ deposition, and at much lower levels [76]. In general, the use of mice as a model of AD presents key limitations. First and foremost, AD is heterogeneous and complex. Wild-type mice do not naturally develop cerebral Aβ deposition and require genetic or biochemical modification to generate plaques. To reduce genetic variance, mice are then inbred and lack genetic heterogeneity. For decades now, the overexpression or knock-in of the human amyloid precursor protein gene carrying FAD-linked mutations has been used to develop mice with cerebral Aβ [172]. APPPS1:E3 and APPPS1:E4 mice co-express human familial AD-related mutations (APPswe and PS1-dE9) [2]. Notably, fewer patients have inherited, in an autosomal dominant manner, familial AD (<1%), and most have the sporadic form (>99%), thereby potentially limiting the translatability of our study. Transgenic mice only reflect amyloid pathogenesis and do not develop neurofibrillary tangles or robust neuron loss. An ideal model would compare human *APOE* alleles in mice that contain both human amyloid and tau pathologies. Finally, our investigations of the effect of the *APOE* genotype are limited by the small sample size of APPPS1:E3 mice. Thus, despite the improved translatability of the APPPS1:E4 mouse model described herein, there remains a great need for heterogeneous and translatable preclinical models that mimic AD. 

We believe that the APPPS1:E4 mouse model is potentially useful, recapitulating the amyloidosis and downstream inflammation and neuritic dystrophy in human AD brains. Prominent plaque and vascular amyloid deposition at 16 months of age demonstrate that this model is well suited for anti-Aβ immunotherapy studies with a focus on vascular-related inflammation.

## 4. Materials and Methods

### 4.1. Animals

The study utilized heterozygous APPswe/PS1dE9 transgenic mice crossbred with homozygous human *APOE3*- and *APOE4*-targeted replacement mice. Our collaborator Dr. Patrick Sullivan generated human *APOE*-targeted replacement mice by replacing the murine *APOE* gene with human *APOE* genomic fragments under regulatory control by the native murine *APOE* promoter and backcrossed them onto a C57BL/6 background [48,49]. Our collaborators Drs. Holtzman and Sadowski crossed these mice with APPswe/PS1dE9 mice [44,45,46,47], the model used in our previous immunotherapy studies [76,173,174], to obtain mice that are heterozygous for APP/PS1dE9 and homozygous for human *APOE3* or *APOE4*. APP/PS1dE9 mice co-express two human genes with familial AD-related mutations, Swedish APP^K594N/M595L^ and Presenilin 1 delta E9 (PS1-dE9; deletion of exon 9), under a mouse prion promoter [44]. APP/PS1de9 mice maintained on a C57BL/6J background have been shown to develop increasing levels of amyloid pathology and behavioral deficits with aging. Our collaborator Dr. Martin Sadowski (NYU) provided the crossed mice for breeding, known as APPPS1:E3 and APPPS1:E4 (David Holtzman’s lab, Washington University). The colony was maintained in a barrier facility with a 12:12 light/dark cycle and housed based on sex and genotype. Food (PicoLab^®^ Rodent Diet 20 5053) and water were available ad libitum. Sample size per age and genotype can be found in Appendix A. All animal studies were approved by the Institutional Animal Care and Use Committees at Harvard Medical School and Brigham and Women’s Hospital.

### 4.2. Euthanasia and Tissue Preparation

Mice were euthanized by excess carbon dioxide inhalation and exsanguinated, and bilateral thoracotomy and final blood collection (obtained by cardiac puncture in the right cardiac ventricle) were performed, followed by transcardial perfusion of phosphate-buffered saline (PBS). Collected blood was stored in Eppendorfs containing 50 mM EDTA and centrifuged at 14,000× *g* at 4 °C for 15 min. Supernatant plasma was stored at −80 °C. Brains were harvested and divided sagittally into two hemispheres. One hemibrain was collected and stored at −80 °C for biochemical analysis. The other hemibrain was fixed overnight at 4 °C in 4% paraformaldehyde (PFA), followed by cryoprotection with 10% and 30% sucrose solutions. Samples were embedded in Optimal Cutting Temperature (OCT) compound (TissueTek, Sakura Finetek USA, Inc., Torrance, CA, USA) for cryosectioning or in UltraPure™ Agarose (Invitrogen, Carlsbad, CA, USA) for vibratome sectioning. Thirty-micron-thick sagittal brain sections were cut using a cryostat (Leica Biosystems, Richmond, IL, USA; CM1850) maintained at –20 °C or a vibrotome (Campden Instruments, Loughborough, Leics., UK; 5100 mz) at room temperature. The free-floating sections were stored in PBS with 0.2% sodium azide at 4 °C.

### 4.3. Immunohistochemistry (IHC), Immunofluorescence (IF), and Histological Staining

In this study, immunohistochemical methods and immunofluorescent staining were applied for sagittal section staining, as previously described by Sun and colleagues [22]. Free-floating sections were incubated with the primary antibodies overnight at 4 °C (see Table 1). Sections were then incubated with the appropriate biotinylated secondary antibodies and developed using Vector ELITE ABC kits (Vector Laboratories, Newark, CA, USA) and 3,3-diaminobenzidine (Sigma-Aldrich, St. Louis, MO, USA) or immunofluorescent-labeled secondary antibodies. The sections were then washed, mounted on glass slides, and cover-slipped with Permount mounting medium (Fischer Scientific, Waltham, MA, USA) or polyvinyl alcohol mounting medium with DABCO. Negative controls for IHC and IF were prepared by either using mouse IgG as the primary antibody or omitting the primary antibody. As described by Liu and colleagues [42], ThS (Sigma-Aldrich) staining was used to detect fibrillar Aβ. For the semi-quantitative analysis of vascular Aβ, non-vascular plaques were manually subtracted in Fiji ImageJ software v1. 54f. For the detection of microhemorrhages, Perl’s hemosiderin staining was performed. 

### 4.4. Biochemical Assays

Mouse brain homogenates were assayed for Aβ, pGlu-3 Aβ, and apoE. Tper-insoluble, guanidine hydrochloride-extracted brain homogenates were extracted from the mice hemibrains. Following the manufacturer’s instructions, we used the Meso Scale Discover (MSD) 96-well multi-spot Human/Rodent (4G8) Aβ Triplex Ultra-Sensitive Assay (K15199E) to simultaneously measure the levels of Aβx-38, Aβx-40, and Aβx-42 and the Immuno-Biological Laboratories (IBL) 96-well Human Amyloid β (N3pE-42) Assay (27716) to measure pGlu-3 Aβ. Under the guidance of our collaborator Dr. Holtzman, apoE levels were assayed utilizing HJ15.6 and HJ15.4b as capture and detection antibodies, respectively [175]. Measurements of lipids and cytokines were taken from blood plasma. Triglyceride (LabAssay Triglyceride, 290-63701), HDL-cholesterol (HDL-Cholesterol E, 997-01301), and total cholesterol (Total Cholesterol-E, 999-02601) concentrations were measured using kits from Wako Diagnostics adapted to half-area 96-well dishes (Corning, 3690). We used the MSD V-PLEX Proinflammatory Panel 1 (mouse) Kit (N05048A-1) to simultaneously measure the levels of ten biomarkers associated with the inflammatory response and immune system regulation.

### 4.5. Image Analysis

Computer-assisted quantification was performed using ImageJ macros, as described before [55,174]. Three to five sagittal images of immunolabeled sections were captured at equidistant levels (~700 μm apart) through each mouse brain using a Zeiss microscope with a motorized stage and a Zeiss AxioCam MRc5 camera, or a Zeiss Axioscan 7 Microscope Slide Scanner (Oberkochen, Germany). All images for one experiment were captured using a constant threshold, and the % area immunoreactivity above the threshold was calculated. Occurrences of vascular amyloid and positive hemosiderin staining were semi-quantitatively counted across full sections. The cerebellum was not included for hemosiderin quantification. 

### 4.6. Statistical Analysis

All values are reported as mean ± SEM. All data were analyzed using the Prism 10.0 statistical software package from GraphPad (San Diego, CA, USA), and outliers (QQplot) were removed. Two-way analyses of variance (ANOVAs) were used to assess the significance between more than two variables. Multiple comparisons were made using either Tukey’s honestly significant difference (HSD) post-test or Šídák’s correction. Unless an interaction involving sex or the main effect of sex was significant, data were collapsed across sex. One-way ANOVA followed by Tukey’s HSD post-test or a non-parametric ANOVA (Kruskal–Wallis) followed by Dunn’s post-test for non-normal data was used to compare multiple groups. Pair-wise comparisons were made using Student’s *t*-test or non-parametric tests such as Alternative Welch’s *t*-test or Mann–Whitney U. In consideration of genotype, sexes were pooled. *p* values less than 0.05 (*p* < 0.05) were considered significant for all tests (* *p* < 0.05, ** *p* < 0.01, *** *p* < 0.001, **** *p* < 0.000). 

## 5. Conclusions

Preclinical studies require suitable animal models that reflect the major characteristics of the disease. Similar to human AD, the presence of a human *APOE4* allele is associated with increased plaque and vascular amyloid deposition with aging. Earlier work has demonstrated the association of vascular amyloid removal by an anti-amyloid antibody with microhemorrhages [83]. With a specific focus on the circumvention of inflammation and microhemorrhages, the prominent Aβ plaque and vascular deposition seen in the APPPS1:E4 mice at 16 months of age support their selection in interventional therapeutic studies. For a preventative study, a timepoint closer to 8 months would be considered reasonable. 

## Figures and Tables

**Figure 1 ijms-25-05754-f001:**
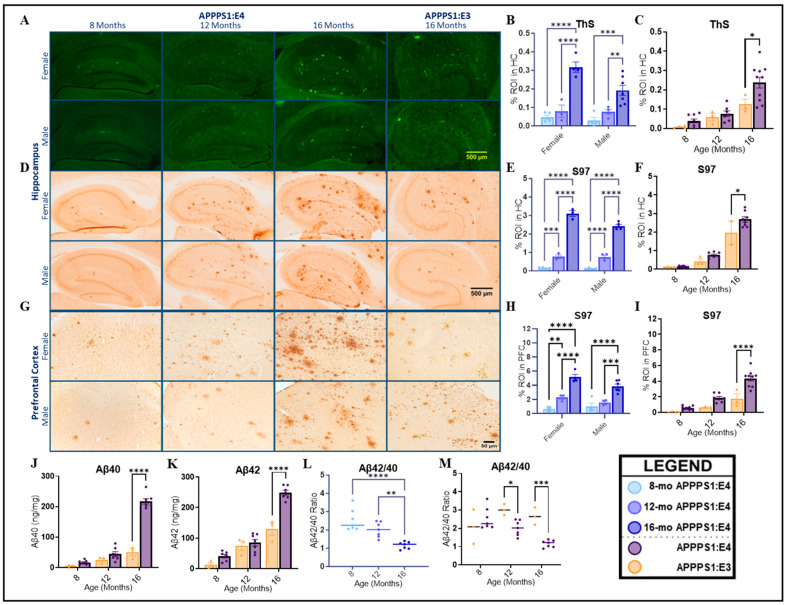
Aβ deposition in hippocampus and prefrontal cortex of APPPS1:E4 and APPPS1:E3 mice. (**A**–**C**) Hemibrain sections from APPPS1:E4 and APPPS1:E3 mice stained with Thioflavin-S for fibrillar Aβ. (**A**) Representative microphotographs of hippocampus. Scale bar: 500 μM. (**B**,**C**) Quantification of fibrillar protein aggregates in hippocampus. (**B**) Analysis of main effect of age in APPPS1:E4 mice (*n* = 3–7 mice/group). (**C**) Analysis of genotype in APPPS1:E4 and APPPS1:E3 mice (*n* = 3–11 mice/group). (**D**–**I**) Hemibrain sections from APPPS1:E4 and APPPS1:E3 mice stained with S97 pAb for general Aβ. (**D**) Representative microphotographs of hippocampus. Scale bar: 500 μM. (**E**,**F**) Quantification of general Aβ deposition in hippocampus. (**E**) Analysis of main effect of age in APPPS1:E4 mice (*n* = 3–5 mice/group). (**F**) Analysis of genotype in APPPS1:E4 and APPPS1:E3 mice (*n* = 2–9 mice/group). (**G**) Representative microphotographs of prefrontal cortex. Scale bar: 50 μM. (**H**,**I**) Quantification of general Aβ deposition in prefrontal cortex. (**H**) Analysis of main effect of age in APPPS1:E4 mice (*n* = 3–6 mice/group). (**I**) Analysis of genotype in APPPS1:E4 and APPPS1:E3 mice (*n* = 3–10 mice/group). (**J**–**M**) Biochemical quantification of common Aβ species with MSD 4G8 Aβ Triplex ELISA on guanidine hydrochloride-extracted whole-hemibrain homogenates from APPPS1:E4 and APPPS1:E3 mice. Analysis of main effect of genotype in APPPS1:E4 and APPPS1:E3 mice on levels of insoluble (**J**) Aβx-40 and (**K**) Aβx-42 (*n* = 3–7 mice/group). (**L**) Analysis of main effect of age in APPPS1:E4 mice on Aβx-42/40 ratio (*n* = 7 mice/group). (**M**) Analysis of main effect of genotype in APPPS1:E4 and APPPS1:E3 mice on Aβx-42/40 ratio (*n* = 7 mice/group). Overall, Aβ deposition significantly increased with age in both hippocampal and cortical areas of brain in APPPS1:E4 mice. Data are presented as mean ± SEM. APPPS1:E4 age effect was analyzed by two-way ANOVA followed by Tukey’s HSD post-test (simple effects within sex). With sexes pooled, genotype effect was analyzed by two-way ANOVA followed by post hoc Sidak’s correction (simple effects within age). * *p* < 0.05; ** *p* < 0.01; *** *p* < 0.001; **** *p* < 0.0001. HC: hippocampus; PFC: prefrontal cortex.

**Figure 2 ijms-25-05754-f002:**
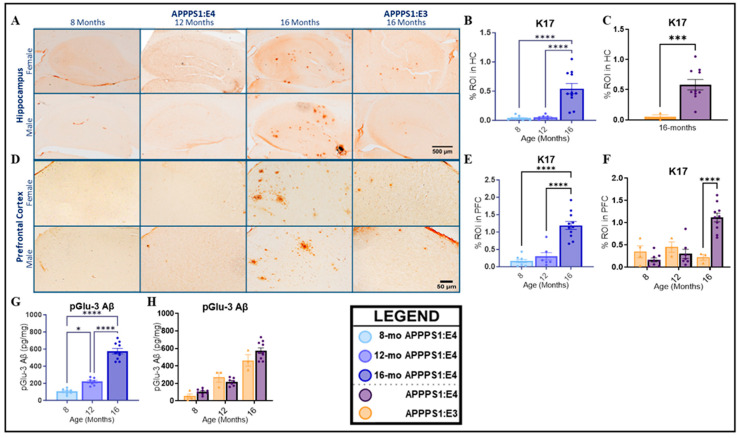
PGlu-3 Aβ deposition and biochemical detection in hippocampus and prefrontal cortex of APPPS1:E4 and APPPS1:E3 mice. (**A**–**F**) Hemibrain sections from APPPS1:E4 and APPPS1:E3 mice stained with K17 IgG2b mAb for pGlu-3-Aβ. (**A**) Representative microphotographs of hippocampus. Scale bar: 500 μM. (**B**,**C**) Quantification in hippocampus. (**B**) Analysis of main effect of age in APPPS1:E4 mice (*n* = 7–11 mice/group). (**C**) Analysis of genotype in 16-month APPPS1:E4 (*n* = 7) and APPPS1:E3 (*n* = 3) mice. (**D**) Representative microphotographs of prefrontal cortex. Scale bar: 50 μM. (**E**,**F**) Quantification in prefrontal cortex. (**E**) Analysis of main effect of age in APPPS1:E4 mice (*n* = 7–11 mice/group). (**F**) Analysis of main effect of genotype in APPPS1:E4 and APPPS1:E3 mice (*n* = 3–10 mice/group). (**G**,**H**) Biochemical quantification of pGlu-3 Aβ levels with IBL N3pE-42 ELISA on guanidine hydrochloride-extracted whole-hemibrain homogenates from APPPS1:E4 and APPPS1:E3 mice. (**G**) Analysis of main effect of age in APPPS1:E4 mice (*n* = 7–11 mice/group). (**H**) Analysis of main effect of genotype in APPPS1:E4 and APPPS1:E3 mice (*n* = 3–10 mice/group). Overall, pGlu-3 Aβ deposition increased significantly at 16 months of age in both hippocampal and cortical areas of brain in female and male APPPS1:E4 mice. Data are presented as mean ± SEM. With sexes pooled, APPPS1:E4 age effect was analyzed with one-way ANOVA followed by post hoc Tukey’s correction. When sex effect was significant within age groups, analysis was conducted by two-way ANOVA followed by post hoc Tukey’s correction. With sexes pooled, genotype effect was analyzed by two-way ANOVA followed by post hoc Sidak’s correction (simple effects within age) or pair-wise comparisons with Alternative Welch’s *t*-test. * *p* < 0.05; *** *p* < 0.001; **** *p* < 0.0001. HC: hippocampus; PFC: prefrontal cortex.

**Figure 3 ijms-25-05754-f003:**
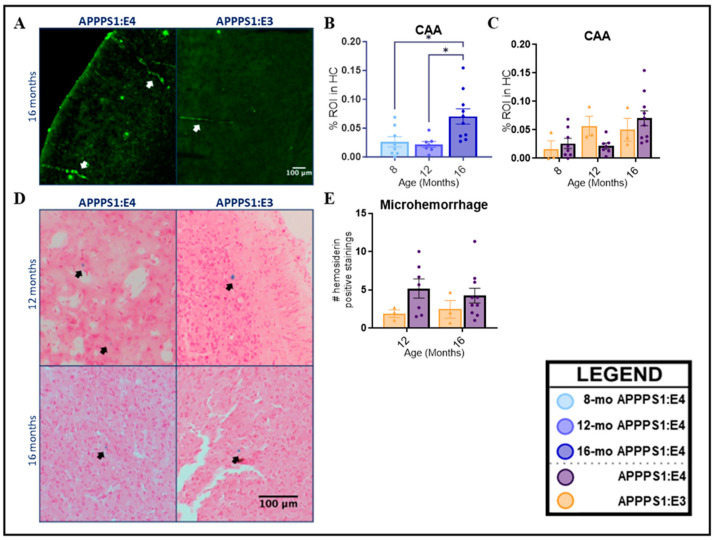
Cerebral amyloid angiopathy (CAA) and microhemorrhage. (**A**–**C**) Hemibrain sections from APPPS1:E4 and APPPS1:E3 mice stained with Thioflavin-S for fibrillar Aβ. (**A**) Representative microphotographs from whole sagittal sections from 16-month-old APPPS1:E4 and APPPS1:E3 mice. Arrowheads indicate long segments of CAA-laden vessels. Scale bar: 100 μM. (**B**,**C**) Semi-quantitative analysis of vascular Aβ deposition. (**B**) Analysis of main effect of age in APPPS1:E4 mice (*n* = 7–10 mice/group). (**C**) Analysis of main effect of genotype in APPPS1:E4 and APPPS1:E3 mice (*n* = 3–10 mice/group). (**D**,**E**) Hemibrain sections from APPPS1:E4 and APPPS1:E3 mice stained with Perls’ Prussian blue for microhemorrhage. (**D**) Representative microphotographs of whole sagittal sections. Arrowheads show hemosiderin-positive staining. Scale bar: 100 μM. (**E**) Semi-quantitative analysis of main effect of genotype in APPPS1:E4 and APPPS1:E3 mice with hemosiderin-positive staining (*n* = 3–10 mice/group). Data are presented as mean ± SEM. With sexes pooled, APPPS1:E4 age effect was analyzed with non-parametric Kruskal–Wallis test followed by Dunn’s post-test. Genotype effect was analyzed by two-way ANOVA followed by post hoc Sidak’s correction (simple effects within age). * *p* < 0.05. HC: hippocampus.

**Figure 4 ijms-25-05754-f004:**
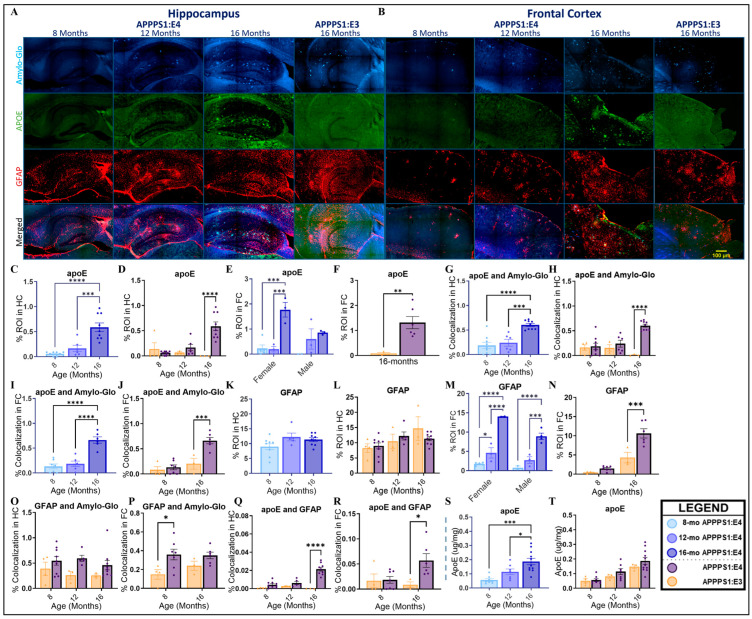
ApoE and astrogliosis. (**A**–**R**). Hemibrain sections from APPPS1:E4 and APPPS1:E3 mice were stained with Amylo-Glo for fibrillar Aβ, anti-apoE mAb for apoE, and anti-GFAP mAb for astrocytes. Representative microphotographs of (**A**) hippocampus (HC; left panel) and (**B**) frontal cortex (FC; right panel). Scale bar: 100 μM. (**C**,**D**) Quantification of apoE immunoreactivity in hippocampus. (**C**) With sexes pooled, analysis of main effect of age in APPPS1:E4 mice (*n* = 6–11 mice/group). (**D**) Analysis of main effect of genotype in APPPS1:E4 and APPPS1:E3 mice (*n* = 3–9 mice/group). (**E,F**) Quantification of apoE immunoreactivity in frontal cortex. (**E**) Analysis of main effect of age in APPPS1:E4 mice (*n* = 1–3 mice/group). (**F**) Analysis of genotype in 16-month APPPS1:E4 and APPPS1:E3 mice (*n* = 3–6 mice/group). (**G**,**H**) Quantification of Amylo-Glo and apoE colocalization in hippocampus, normalized by positive staining of Amylo-Glo. (**G**) With sexes pooled, analysis of main effect of age in APPPS1:E4 mice (*n* = 6–11 mice/group). (**H**) Analysis of main effect of genotype in APPPS1:E4 and APPPS1:E3 mice (*n* = 3–9 mice/group). (**I**,**J**) Quantification of Amylo-Glo and apoE colocalization in frontal cortex, normalized by immunoreactivity of Amylo-Glo. (**I**) With sexes pooled, analysis of main effect of age in APPPS1:E4 mice (*n* = 6–7 mice/group). (**J**) Analysis of main effect of genotype in 8- and 16-month-old APPPS1:E4 and APPPS1:E3 mice (*n* = 3–7 mice/group). (**K**,**L**) Quantification of GFAP immunoreactivity in hippocampus. (**K**) With sexes pooled, analysis of main effect of age in APPPS1:E4 mice (*n* = 5–9 mice/group). (**L**) Analysis of main effect of genotype in APPPS1:E4 and APPPS1:E3 mice (*n* = 3–9 mice/group). (**M**,**N**) Quantification of GFAP immunoreactivity in frontal cortex. (**M**) Analysis of main effect of age in APPPS1:E4 mice (*n* = 2–5 mice/group). (**N**) Analysis of main effect of genotype in 8- and 16-month APPPS1:E4 and APPPS1:E3 mice (*n* = 3–7 mice/group). (**O**) Quantification of Amylo-Glo and GFAP colocalization in hippocampus, normalized by positive staining of Amylo-Glo. Analysis of main effect of genotype in APPPS1:E4 and APPPS1:E3 mice (*n* = 3–9 mice/group). (**P**) Quantification of Amylo-Glo and GFAP colocalization in frontal cortex, normalized by immunoreactivity of Amylo-Glo. Analysis of main effect of genotype in 8- and 16-month APPPS1:E4 and APPPS1:E3 mice (*n* = 3–7 mice/group). (**Q**) Quantification of apoE and GFAP colocalization in hippocampus, normalized by immunoreactivity of GFAP. Analysis of main effect of genotype in APPPS1:E4 and APPPS1:E3 mice (*n* = 3–9 mice/group). (**R**) Quantification of apoE and GFAP colocalization in frontal cortex, normalized by immunoreactivity of GFAP. Analysis of main effect of genotype in 8- and 16-month APPPS1:E4 and APPPS1:E3 mice (*n* = 3–7 mice/group). (**S**,**T**) Biochemical quantification of apoE levels following Liao et al.’s 2015 protocol on guanidine hydrochloride-extracted whole-hemibrain homogenates from APPPS1:E4 and APPPS1:E3 mice. (**S**) Analysis of main effect of age in APPPS1:E4 mice (*n* = 7–11 mice/group). (**T**) Analysis of main effect of genotype in APPPS1:E4 and APPPS1:E3 mice (*n* = 3–11 mice/group). Data are presented as mean ± SEM. Colocalization data are presented in arbitrary units. APPPS1:E4 age effect was analyzed by one-way ANOVA followed by post hoc Tukey’s analysis or two-way ANOVA followed by Tukey’s HSD post-test (simple effects within sex). With sexes pooled, genotype effect was analyzed by two-way ANOVA followed by post hoc Sidak’s correction (simple effects within age) or pair-wise comparisons with Alternative Welch’s *t*-test. Due to low n-value, frontal cortex data were not evaluated for 12-month-old APPPS1:E3 mice. * *p* < 0.05; ** *p* < 0.01; *** *p* < 0.001; **** *p* < 0.0001. HC: hippocampus; FC: frontal cortex.

**Figure 5 ijms-25-05754-f005:**
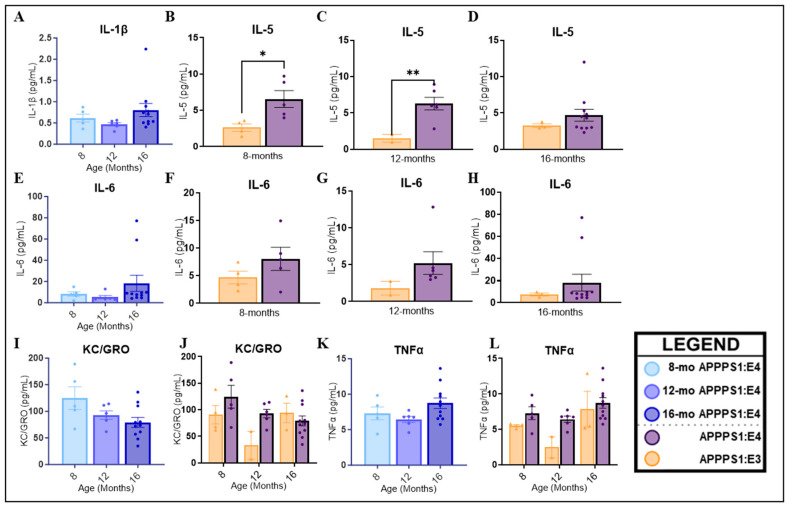
Inflammatory biomarkers in mouse plasma. (**A**–**D**) Quantification of IL-1β levels. (**A**) Analysis of main effect of age in APPPS1:E4 mice (*n* = 6–11 mice/group). Analysis of genotype in APPPS1:E4 and APPPS1:E3 mice at (**B**) 8 months (*n* = 4–5 mice/group), (**C**) 12 months (*n* = 2–6 mice/group), and (**D**) 16 months (*n* = 3–11 mice/group). (**E**–**H**) Quantification of IL-6 levels. (**E**) Analysis of main effect of age in APPPS1:E4 mice (*n* = 6–11 mice/group). Analysis of genotype in APPPS1:E4 and APPPS1:E3 mice at (F) 8 months (*n* = 4–5 mice/group), (**G**) 12 months (*n* = 2–6 mice/group), and (**H**) 16 months (*n* = 3–11 mice/group). (**I**,**J**) Quantification of KC/GRO (IL-8-related protein in rodents) levels. (**I**) Analysis of main effect of age in APPPS1:E4 mice (*n* = 6–11 mice/group). (**J**) Analysis of main effect of genotype in APPPS1:E4 and APPPS1:E3 mice (*n* = 2–11 mice/group). (**K**,**L**). Quantification of TNF-α levels. (**K**) Analysis of main effect of age in APPPS1:E4 mice (*n* = 6–11 mice/group). (**L**) Analysis of main effect of genotype in APPPS1:E4 and APPPS1:E3 mice (*n* = 2–11 mice/group). Data are presented as mean ± SEM. With sexes pooled, APPPS1:E4 age effect was analyzed with one-way ANOVA followed by post hoc Tukey’s correction or non-parametric ANOVA (Kruskal–Wallis) followed by Dunn’s post-test. With sexes pooled, genotype effect was analyzed by two-way ANOVA followed by post hoc Sidak’s correction (simple effects within age) or pair-wise comparisons with Alternative Welch’s *t*-test. * *p* < 0.05; ** *p* < 0.01.

**Figure 6 ijms-25-05754-f006:**
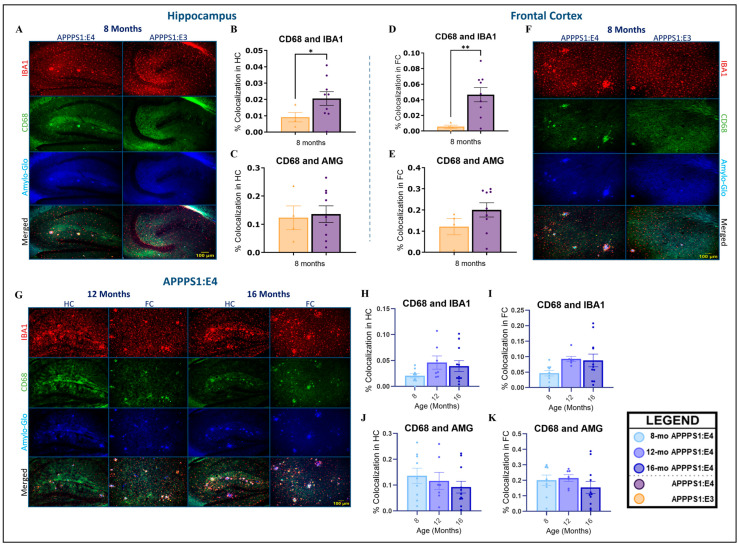
The microglial response in the hippocampus and prefrontal cortex of APPPS1:E4 and APPPS1:E3 mice throughout aging. (**A**) IBA1, CD68, and Amylo-Glo staining in the hippocampal regions of APPPS1:E4 and APPPS1:E3 mice at 8 months of age. Scale bar: 100 μM. (**B**–**E**) Phagocytic microglia in the hippocampus (**B**) and frontal cortex (**D**) of APPPS1:E4 and APPPS1:E3 mice at 8 months of age were quantified by the % colocalization area of CD68 and IBA1 staining. Plaque-associated lysosomal activity in the hippocampus (**C**) and frontal cortex (**E**) of APPPS1:E4 and APPPS1:E3 mice at 8 months of age was quantified by the % colocalization area of CD68 and Amylo-Glo staining (*n* = 4–9 mice/genotype group). (**F**) IBA1, CD68, and Amylo-Glo staining in the frontal cortex of APPPS1:E4 and APPPS1:E3 mice at 8 months of age. Scale bar: 100 μM. (**G**) IBA1, CD68, and Amylo-Glo staining in the hippocampus and frontal cortex of APPPS1:E4 mice at 12 and 16 months of age. Scale bar: 100 μM. (**H**–**K**) The quantification of phagocytic microgliosis (**H**,**I**) and plaque-associated lysosomal activity (**J**,**K**) in the hippocampus (**H**,**J**) and frontal cortex (**I**,**K**) of APPPS1:E4 mice, indicating an aging effect. (*n* = 7–11 mice/age group). Overall, a significant genotype effect is only observed at 8 months of age, while the microglia of APPPS1:E4 mice had an impaired response to plaque deposition during aging. Data are presented as mean ± SEM. Colocalization data are presented in arbitrary units. The genotype effect was analyzed by Alternative Welch’s test. The age effect was analyzed by one-way ANOVA followed by post hoc Tukey’s correction. * *p* < 0.05; ** *p* < 0.01. HC: hippocampus; FC: frontal cortex; AMG: Amylo-Glo.

**Figure 7 ijms-25-05754-f007:**
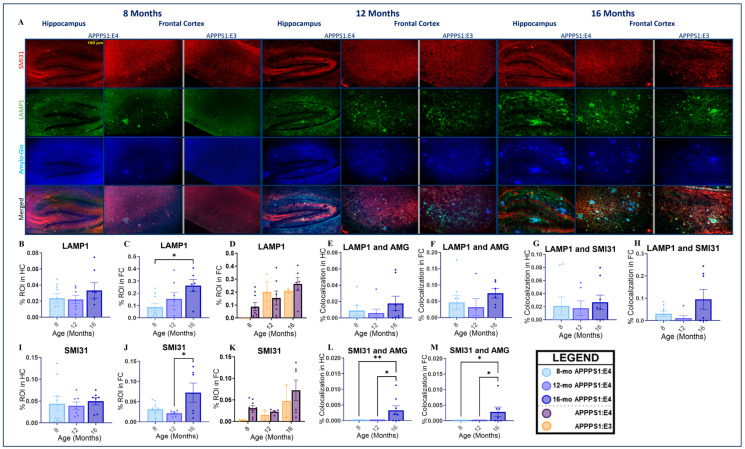
Lysosomal dysfunction and neuritic dystrophy in the hippocampus and frontal cortex of APPPS1:E4 and APPPS1:E3 mice. (**A**) SMI-31, LAMP1, and Amylo-Glo staining in the hippocampus and frontal cortex of APPPS1:E4 mice and the frontal cortex of APPPS1:E3 mice at 8 months of age (left panel), at 12 months of age (middle panel), and at 16 months of age (right panel). Scale bar: 100 μM. (**B**–**D**) Lysosomal accumulation in the hippocampus (**B**) and frontal cortex (**C**,**D**) of APPPS1:E4 and APPPS1:E3 mice was quantified by the % area of LAMP1^+^ immunoreactivity and indicated an effect of aging in APPPS1:E4 mice (**B**,**C**) (*n* = 5–8 mice/age group) and a genotype effect in the frontal cortex of APPPS1:E4 and APPPS1:E3 mice at the three different age points (**D**) (*n* = 2–8 mice/genotype group). (**E**,**F**) Plaque-associated LAMP1^+^ immunoreactivity in the hippocampus (**E**) and frontal cortex (**F**) of APPPS1:E4 mice was quantified by the % colocalization area of LAMP1 and Amylo-Glo staining and indicated an effect of aging in APPPS1:E4 mice (**E**,**F**) (*n* = 5–8 mice/age group). (**G**,**H**) Proteasomal dysfunction in neurites in the hippocampus (**G**) and frontal cortex (**H**) of APPPS1:E4 mice was quantified by the % colocalization area of LAMP1 and SMI31 staining and indicated an aging effect in both brain areas (**G**,**H**) (*n* = 5–8 mice/age group). (**I**–**K**) The accumulation of dystrophic neurites in the hippocampus (**I**) and frontal cortex (**J**,**K**) of APPPS1:E4 and APPPS1:E3 mice was quantified by the % area of SMI31+ immunoreactivity and indicated an effect of aging in APPPS1:E4 mice (**I**,**J**) (n =5–8 mice/age group) and a genotype effect in the frontal cortex of APPPS1:E4 and APPPS1:E3 mice at the three different age points (**K**) (*n* = 2–8 mice/genotype group). (**L**,**M**) Plaque-associated neuritic dystrophy was quantified by the % colocalization area of SMI-31 and Amylo-Glo staining and showed an effect of age in the hippocampus (**I**) and frontal cortex (**J**) in APPPS1:E4 mice (*n* = 5–8 mice/age group). Data are presented as mean ± SEM. Colocalization data are presented in arbitrary units. The age effect was analyzed by one-way ANOVA followed by Tukey’s post hoc correction or by Kruskal–Wallis followed by Dunn’s post hoc correction. The genotype effect was analyzed by two-way ANOVA followed by post hoc Sidak’s correction or by Welch’s test. * *p* < 0.05; ** *p* < 0. 01. HC: hippocampus; FC: frontal cortex; AMG: Amylo-Glo.

**Table 1 ijms-25-05754-t001:** Antibodies for immunohistochemistry and histology reagents. The antibodies are monoclonal or polyclonal affinity-purified mouse (M), rabbit (Rb), or rat (Rt). Aβ: amyloid beta; Aβ-pE(3): Pyroglutamylate-3 amyloid beta; ApoE: apolipoprotein E; CD68; cluster of differentiation 68; GFAP: glial fibrillary acidic protein; Iba1: ionized calcium-binding adaptor molecule 1; LAMP-1: Lysosome-Associated Membrane Protein 1; SMI-31: Sternberger monoclonals incorporated-31.

Name	Catalog #	Host/Type	Source	Marker
Amylo-Glo	6264-656	N/A	VWR International (Radnor, PA, USA)	Aβ plaques
ApoE	13366	Rb-monoclonal	Cell Signaling (Danvers, MA, USA)	apoE
CD68	MCA341GA	Rt-monoclonal	BIO-RAD (Hercules, CA, USA)	Monocytes/macrophages
GFAP	G3893	M-monoclonal	Millipore Sigma (Burlington, MA, USA)	Astrogliosis
IBA1	019-19741	Rb-polyclonal	FUJIFILM Wako Chemicals (Osaka, Japan)	Microglia/macrophages
K17	N/A	M-monoclonal	Stephen Schilling (Probiodrug AG/Vivoryon Therapeutics NV, Halle, Germany)	Aβ-pE(3)
LAMP-1	25245	Rt-monoclonal	Abcam (Cambridge, UK)	Lysosomes
Potassium ferrocyanide	P9387	N/A	Sigma Aldrich (St. Louis, MO, USA)	Hemosiderin/microhemorrhages
S97	N/A	Rb-polyclonal	Dr. Dominic Walsh, BWH (Boston, MA, USA)	General Aβ
SMI-31	801601	M-monoclonal	BioLegend (San Diego, CA, USA)	Extensively phosphorylated neurofilaments
Thioflavin-S	T1892	N/A	Sigma Aldrich (St. Louis, MO, USA)	Fibrillar Aβ plaques and vascular amyloid

## Data Availability

Data supporting the findings are available on request from the corresponding author.

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
