# Peer review of "Temporal Characterization of the Amyloidogenic APPswe/PS1dE9;hAPOE4 Mouse Model of Alzheimer’s Disease"

_ijms, 2024, doi:10.3390/ijms25115754_

Round 1

Reviewer 1 Report

Comments and Suggestions for Authors

The manuscript you have submitted for review is one of the best I have reviewed. The study design is flawless, and the quality of the data you have generated is of a high standard. Most of the important points that I can think of have already been covered in the manuscript, and everything in it is relevant.

I have only minor comments that can be easily addressed:

  1. It would be interesting to add a graphical abstract summarizing the findings.

  2. Regarding CAA-microhemorrhage, I would suggest studying the integrity of the BBB using Claudin 5 and ABCB1-LRP1.

  3. Besides the high quality of the data, I would suggest choosing different colors for the graphs, as it might be confusing when the colors are too similar.

Thank you

Author Response

We sincerely thank the reviewer for their enthusiasm regarding our paper and their valuable feedback. 

  • It would be interesting to add a graphical abstract summarizing the findings.
    • Response: Great suggestion. We have inputted!
  • Regarding CAA-microhemorrhage, I would suggest studying the integrity of the BBB using Claudin 5 and ABCB1-LRP1.
    • Response: Thank you for this excellent suggestion. We are pursuing this as part of another study, which will be a separate manuscript.
  • Besides the high quality of the data, I would suggest choosing different colors for the graphs, as it might be confusing when the colors are too similar.
    • Response: Thank you, we have modified the graph colour scheme for analyses of APOE genotype (APPPS1:E3 and APPPS1:E4). We agree that it is much more clear now.

Reviewer 2 Report

Comments and Suggestions for Authors

This study dealt with the establishment of mice model regarding AD with three different ages, by using APPswe/PS1dE9 by contrasting with APOE4 and APOE3 from the histrogical to biochemical viewpoints. The analysis is as far as systematic enough to elucidate a valuable conclusion to a certain extent. Furthermore, the authors deepened the results at the Discussion section by consulting more than 150 articles. After a revision of Figure 6  broken at the right side, the present work should be discussed by general readers.

Author Response

We sincerely thank the reviewer for their enthusiasm regarding our paper and their valuable feedback.

  • After a revision of Figure 6 broken at the right side, the present work should be discussed by general readers
    • Response: Thank you. Realized this formatting issue was a result of the upload process. Figure 6 moved to the center, and the right side is visible now.

Reviewer 3 Report

Comments and Suggestions for Authors

The manuscript by Grenon et al named” TEMPORAL CHARACTERIZATION OF THE AMYLOIDO-2 GENIC APP/PS1dE9;hAPOE4 MOUSE MODEL OF ALZHEIMER’S DISEASE” describes the author's study of the importance of human APOE phenotype in popular transgenic mice model of Alzheimer's, for the development of different forms of Aβ deposition in this model.  The authors compare the APPswe/PS1dE9 transgenic mouse model bearing the human APOE4 gene, with the same APP/PS1 mice bearing a human APOE3 gene in both female and male mice of different ages. Proteins, coded by APOE ε3 differ from APOE ε4 in only one amino acid, all brain APOE are lipid carriers participating in the transport of lipids and other lipid-soluble molecules between brain cells and vessels. Authors have found that in transgenic mice male and female  APOE ε4 carriers showed elevated Aβ, ApoE, reactive astrocytes, pro-inflammatory cytokines, microglial response, and neuritic dystrophy compared to APOE ε3 carriers at different ages.

The experiments are clear and well-performed, and the description of the experimental procedure is clear. Overall, the manuscript is well-written, and this reviewer liked the author's work. It is written in good English and contains practically no flow visible to this reviewer.  All figures (including supplemental ones) and the statistics are well done.

The only moment not well described is why authors used this popular mouse model for their study and the model description. It is known that rodent beta-amyloid does not form insoluble fibrils (Jankowsky et al ,2007) and to obtain not a temporal, but Alsheimer-like accumulation of Aβ the rodents need to have inserted human APP genes and machinery to process this human APP. According to this, a mouse model was constructed using a human APP bearing Swedish mutation (APPswe) and a mutated human presenilin complex (PS1dE9), which was inserted in the genome using a neuronal promoter (Jankowsky et al, 2001). Similarly, APOE variants use the same promoter, thus forcing neurons to produce APOE (normally, it is mainly expressed in astrocytes) and human-beta-amyloid. In this model, neurons are the main producers of beta-amyloid and APOE variants, and their production is intrinsically correlated because it use the same promoter. It may give the wrong answers if one tries to study how APOE affects beta-amyloid production.

 In humans, APOE4 is associated with low levels of ApoE in the brain, CSF, and plasma in healthy controls and AD patients compared to APOE3 (Riddell et al, 2008).  In your data on transgenic mice, on the contrary, it is elevated. Why?

Also, in your study, the main effect of APOE variants became visible in aged mice, while in humans, the main effect is mostly seen in early onset. Moreover,  it was shown that APOE variants in humans affect familial Alzheimer’s (like Swedish) 3 times more, than other subtypes of Alzheimer's (Jia et al, 2020). Because the used model represents familiar mutation, how it is related to other more common subtypes of this human disease?

Generally speaking, this mice model is good for observing amyloid plaques in the brain, but not for correlation studies. Even beta-amyloid plaques in this model are chimeric, containing human amyloid core, and mice amyloid periphery (van Groen et al, 2006). Most probably, APOE carriers in this model work also in the wrong direction, because in this model they transport beta-amyloid (attached to lipids) from neurons to blood vessels, while if beta-amyloid is produced in the periphery (like many authors propose) APOE have to transport Aβ from blood vessels to brain cells (Lam et al.2021).

All this suggests that authors need to better describe the used model and why they think it shares the same effects of APOE variants as human disease, especially the effects of beta-amyloid abnormalities on blood vessels (ARIA).

After this minor addition, the work may be published.   

Literature;

Riddell DR, et al. Impact of apolipoprotein E (ApoE) polymorphism on brain ApoE levels. J Neurosci. 2008;28(45):11445–53.

Jankowsky JL, et al. Rodent A beta modulates the solubility and distribution of amyloid deposits in transgenic mice. J Biol Chem (2007) 282(31):22707–20. doi: 10.1074/jbc.M611050200

Jankowsky, J.L., et al., Co-expression of multiple transgenes in mouse CNS: a comparison of strategies. Biomol Eng, 2001. 17(6): p. 157-1061 65.

Jia et al, The APOE ε4 exerts differential effects on familial and other subtypes of Alzheimer's disease. Alzheimers Dement. 2020 Dec;16(12):1613-1623. doi: 10.1002/alz.12153.

van Groen T, Kiliaan AJ, Kadish I. Deposition of mouse amyloid beta in human APP/PS1 double and single AD model transgenic mice. Neurobiol Dis. 2006 Sep;23(3):653-62. doi: 10.1016/j.nbd.2006.05.010.

Lam V, et al. Synthesis of human amyloid restricted to liver results in an Alzheimer disease-like neurodegenerative phenotype. PLoS Biol. 2021 Sep 14;19(9):e3001358. doi: 10.1371/journal.pbio.3001358.

Author Response

We sincerely thank the reviewer for their enthusiasm regarding our paper and their valuable feedback.

  • The only moment not well described is why authors used this popular mouse model for their study
    • Response: We have added further rationale (introduction, methods, discussion, and conclusion) for the selection of this mouse model as our specific research question revolves around APOE4 influence on Aβ deposition, including vascular amyloid.
  • Similarly, APOE variants use the same promoter, thus forcing neurons to produce APOE (normally, it is mainly expressed in astrocytes) and human-beta-amyloid. In this model, neurons are the main producers of beta-amyloid and APOE variants, and their production is intrinsically correlated because it use the same promoter.
    • Response: We are unable to find any publications showing that the APOE targeted replacement mice express human APOE in neurons. According to Sullivan 2004; the human APOE targeted replacement mice “showed a predominantly glial (astrocytic) pattern of ApoE expression in both grey and white matter”.
  • In humans, APOE4 is associated with low levels of ApoE in the brain, CSF, and plasma in healthy controls and AD patients compared to APOE3 (Riddell et al, 2008).  In your data on transgenic mice, on the contrary, it is elevated.
    • Response: Thank you for pointing out that our discussion of these results required additional comments. We updated the discussion section on APOE levels and believe it is now more of a thorough overview. We added additional original primary references of studies measuring the directionality of APOE4 carriers and differences in APOE levels. There is evidence, in AD patients and mouse models, for both the increase and decrease of APOE levels. Regarding the inconsistency of the direction of the effect of APOE4 on ApoE protein levels, we further expanded on the caveats between these studies.
  • Also, in your study, the main effect of APOE variants became visible in aged mice, while in humans, the main effect is mostly seen in early onset.
    • Response: Thank you. Added sentences to the discussion expanding on differences in APOE effects in mice and humans. However, it should be noted that aging plays a major role in APOE effects in both humans and mice.
  • It was shown that APOE variants in humans affect familial Alzheimer’s (like Swedish) 3 times more, than other subtypes of Alzheimer's (Jia et al, 2020). Because the used model represents familiar mutation, how it is related to other more common subtypes of this human disease?
    • Response: Thank you. We have added a few sentences in the limitations section of the discussion to acknowledge that this mouse model carries 2 FAD linked mutations, and, therefore, one limit of this study is that it does not necessarily represent patients with sporadic AD. Unfortunately to date, mouse models expressing human wild type APP do not develop amyloid pathology and therefore, it is necessary to express familial AD-linked mutations (in either APP or presenilin).
  • Generally speaking, this mice model is good for observing amyloid plaques in the brain, but not for correlation studies. Even beta-amyloid plaques in this model are chimeric, containing human amyloid core, and mice amyloid periphery (van Groen et al, 2006). Most probably, APOE carriers in this model work also in the wrong direction, because in this model they transport beta-amyloid (attached to lipids) from neurons to blood vessels, while if beta-amyloid is produced in the periphery (like many authors propose) APOE have to transport Aβ from blood vessels to brain cells (Lam et al.2021).
    • Response: Thank you. We have added explanations of the limitations of this mouse model to the discussion. We believe our data supports the use of the APPPS1:E4 mouse model in answering specific research questions, e.g. vascular side effects associated with anti-amyloid immunotherapy. Therefore, we do not feel that the Lam et al., 2021 paper showing a possible periphery to CNS delivery of beta-amyloid is relevant here and therefore, we did not include it.
  • All this suggests that authors need to better describe the used model and why they think it shares the same effects of APOE variants as human disease, especially the effects of beta-amyloid abnormalities on blood vessels (ARIA).
    • Response: Thank you for this suggestion. As mentioned above, we have added additional information to the Introduction to provide a stronger rationale for the selection of this mouse model for our studies.

Sullivan, P. M., Mace, B. E., Maeda, N., & Schmechel, D. E. (2004). Marked regional differences of brain human apolipoprotein E expression in targeted replacement mice. Neuroscience124(4), 725–733. https://doi.org/10.1016/j.neuroscience.2003.10.011